# Vocalization during agonistic encounter in Mongolian gerbils: Impact of sexual experience

**Takafumi Furuyama[1]\*, Takafumi Shigeyama[2], Munenori Ono[1], Sachiko Yamaki[1], Kohta I. Kobayasi[2], Nobuo Kato[1], Ryo Yamamoto[1]\***

**1** Department of Physiology, Kanazawa Medical University, Ishikawa, Japan, **2** Graduate School of Life and Medical Sciences, Doshisha University, Kyoto, Japan

\* ryamamot@kanazawa-med.ac.jp (RY); tfuruyam@kanazawa-med.ac.jp (TF)

## Abstract

Behaviors and vocalizations associated with aggression are essential for animals to survive, reproduce, and organize social hierarchy. Mongolian gerbils (*Meriones unguiculatus*) are highly aggressive and frequently emit calls. We took advantage of these features to study the relationship between vocalizations and aggressive behaviors in virgin and sexually experienced male and female Mongolian gerbils through the same-sex resident-intruder test. Both sexes of resident gerbils exhibited aggressive responses toward intruders. Multiparous females exhibited the most aggressive responses among the four groups. We also confirmed two groups of vocalizations during the encounters: high-frequency (>24.6 kHz) and low-frequency (<24.6 kHz). At the timing of high-frequency vocalizations observed during the tests, the vast majority (96.2%) of the behavioral interactions were non-agonistic. While, at the timing of low-frequency vocalizations observed during the tests, around half (45%) of the behavioral interactions were agonistic. Low-frequency vocalizations were observed mainly during encounters in which multiparous females were involved. These results suggest that high- and low-frequency vocalizations relate to non-agonistic and agonistic interactions, respectively. In addition to affecting aggressive behavior, sexual experience also affects vocalization during encounters. These findings provide new insights into the modulatory effects of sex and sexual experience on vocalizations during agonistic encounters.

## Introduction

In most species, aggression is an adaptive response to increase survival and reproduction rates. Territorial aggression is one of the most common aggressive behaviors displayed by rodents [1–3], and it consists of multiple complex behaviors, such as biting, boxing, and chasing. In addition, unique vocalizations associated with territorial aggression have been reported in males of many rodents [4]. As vocal communication is important for the survival and formation of a colony, rodents emit vocalizations in various behavioral contexts, including aggressive encounters [5,6]. Furthermore, past experiences have also been found to modulate

**Funding:** This work was supported partly by KAKENHI Grants JP18KK0468 and JP21K07489 (to R. Y.), JP19K09918 (to M. O.), JP21K11251 (S. Y.) JP21H03469 (to K. I. K.), JP17H02223 (to N. K.). JP19K16192 (to T. F.) Kawano Masanori Memorial Public Interest Incorporated Foundation for Promotion of Pediatrics (to T.F.) The Kawai Foundation for Sound Technology & Music (to T. F.) Casio Science Foundation (to T.F.) No sponsors or funders play any role in the study design, data collection and analysis, decision to publish, or preparation of the manuscript.

**Competing interests:** Authors declare no conflict of interest.

vocalizations in rodents [7–9]. However, it is not well known how behavioral contexts and past experiences modulate vocalizations during aggressive encounters.

The Mongolian gerbil (*Meriones unguiculatus*) is a highly social and territorial rodent species [10–12]. These animals exhibit elevated territorial aggressiveness by which they attack, or in some cases even kill, unfamiliar conspecifics [13]. As it is well known, laboratory rodents such as mice or rats rarely exhibit this extent of aggression to kill the other conspecifics [3,6]. Aggressiveness in Mongolian gerbils has been observed during same-sex and opposite-sex interactions [14]. Furthermore, sexually naïve males and even females exhibit aggressive behavior toward unfamiliar same-sex conspecifics [15,16], which demonstrates that aggressiveness toward unfamiliar conspecifics is a prominent feature of both sexes of Mongolian gerbils. In addition, sexually experienced gerbils are more aggressive than virgin subjects [13,17].

Although Mongolian gerbils are aggressive toward unfamiliar conspecifics, they communicate with each other, even with unfamiliar conspecifics, frequently through vocalization. The vocalizations of Mongolian gerbils have been confirmed and investigated in various behavioral contexts [17–19], including encounters with conspecifics [20], mother-infant interactions [21–23], and mating [24]. Moreover, the call rates of these vocalizations in Mongolian gerbils are sexually dimorphic [24,25].

Given that both sexes are aggressive and emit calls frequently, Mongolian gerbils are one of the best laboratory animals to investigate vocalizations during agonistic encounters. Although several studies cited in the previous paragraph have reported aggression-specific vocalizations, it remains unclear how sex and sexual experience affect the call rate of various vocalizations during agonistic encounters in Mongolian gerbils. Thus, we analyzed vocalizations between four different pair-tests (virgin male–virgin male; experienced male–virgin male; virgin female–virgin female; multiparous female–virgin female) of Mongolian gerbils through the same-sex resident-intruder test, to clarify the difference of vocalizations and relation between vocalizations and agonistic interactions.

## Materials and methods

All experiments were conducted in accordance with the guidelines and protocols approved by the Animal Care Committee of Kanazawa Medical University (2019–22, 2022–6) and the Animal Experimental Committee of Doshisha University (A21052-1).

### Subjects

Fifty-two Mongolian gerbils (24 males and 28 females, aged 3–6 months; purchased from Sankyo Lab Service, Tokyo, Japan) were used. Five males were sexually experienced, and seven females were multiparous. These gerbils were mated at our facility. The other gerbils were virgins. All gerbils were bred and maintained in the laboratory at 22–23°C, and approximately 50% humidity. Two to four animals were housed together in plexiglass cages (20 cm × 40 cm × 17 cm; white paper beddings) under a 12 h light/dark cycle. Food and water were available *ad libitum*.

### Same-sex resident-intruder test

We conducted 26 resident-intruder tests to assess the extent of territorial aggressiveness in the gerbils. Each subject was singly housed in a plexiglass cage (20 cm × 40 cm × 17 cm; resident cage) for at least one week before undergoing the resident-intruder test. The residents were divided into four groups: virgin females (VF; n = 7), multiparous females (MF; n = 7), virgin males (VM; n = 7), and experienced males (EM; n = 5). All tests were performed

during the light cycle of the day. At the test, each resident's home cage was placed in a sound-attenuated room one by one. Three minutes later, an intruder was introduced into the resident's cage. All intruders were virgin, of the same sex as the resident and unfamiliar to the resident (12 VMs and 14 VFs). Each subject (resident and intruder) underwent the resident-intruder test for a single time. Most of the residents (24/26) expressed aggressive behaviors toward the intruder within several minutes. Aggressive behaviors were initiated by residents in the vast majority (20/26) of tests, thus basically we focused on the behaviors of residents in this study. Behavioral interactions between the resident and intruder were recorded using a CMOS camera (D435, Intel Corporation, Santa Clara, CA, USA) placed in front of the cage. For behavioral analysis, we converted the frame rate of the recorded files to 10/s, and visually identified the types of behaviors of the resident and intruder by following the classification described in a previous report [17] with a small modification. The behaviors were classified as follows by observers manually: alert posture, ano-genital sniff, approach, attack, boxing, chase, dig, explore, fight, flee, jump, move away, nasal sniff, push, self-groom, stop moving, and watch. Details of each behavior are shown in Table 1. 'attack', 'boxing', 'chase', 'fight', 'flee', and 'push' were scored as aggressive/agonistic behaviors. The rest was scored as non-agonistic interaction. The latency of aggressive behavior (start time of the first aggressive behavior) and the total duration of aggressive behavior were recorded. The total duration of aggressive behavior per minute was defined as 'normalized duration of aggressive behavior'. The intruder was removed from the resident cage after 10 min of behavioral testing. If no aggressive behavior was observed within 10 min, the latency of aggressive behavior was regarded as 600 s, and the duration of aggressive behavior was scored as 0 s for the analysis. The tests were halted immediately when aggressive behaviors escalated to physical injuries. In this study, 3 tests (three female pairs) were halted at 3–5 min. These cases were included in this study, by using each index for behaviors and vocalizations normalized by time (duration or number of calls per minute).

**Table 1. Descriptions of behaviors that were scored.**

| Behaviors | Description |
| --- | --- |
| Attack | One animal biting the another |
| Fight | Two animals gripping each other's flanks, biting, and rolling over |
| Flee | Running away from the another |
| Boxing | Two animals facing each other with physical contact |
| Chase | One animal running after another more than one body length |
| Push | Using paws or body to make another move away |
| Jump | Pushing itself off the ground with the hind legs in a vertical movement |
| Alert | Standing on the hind legs alone and watching the surroundings |
| Watch | Both animals keeping motionless, face-to-face at a close distance |
| Move away | Making a distance from the another |
| Dig | Removing the beddings with its front paws moving quickly back and forth |
| Grooming | Licking, nibbling, and scratching of one's own body |
| Explore | Actively moving around the cage like investigating |
| Stop | Staying still |
| Approach | Moving nearer to another within one body length |
| Nasal sniff | Nose-to-nose contact |
| Ano-genital sniff | Nose contact to the ano-genital region of another animal |

Descriptions of each behavior, following the scoring in the previous reports [17,26] with small modifications.

## Recording and analysis of vocalizations

A recording microphone (Ultrasound Microphone CMPA-P48/CM16 SN34, Avisoft Bioacoustics, Berlin, Germany) was placed 20 cm above the top of the cage, and vocalizations were recorded using a sound card (UltraSoundGate, 116Hb, Avisoft Bioacoustics, Berlin, Germany) at a sample rate of 125 kHz at 16 bits/sample. To identify calls, Adobe Audition software (Adobe, CA, US) was used. Each call was detected and cut out from the audio files manually by experimenters. All recorded calls with a signal-to-noise ratio higher than 10 dB were used for offline analysis and classified into different syllable types. This analysis was based on a 512-point fast Fourier transform (Hamming window) with an 80% overlap. Spectrograms were calculated using a MATLAB custom program at a frequency resolution of 244 Hz and a temporal resolution of 0.82 ms. Call envelopes were obtained from the original waveforms. The spectro-temporal parameters of vocalizations were quantified using the following parameters: total duration, frequency of fundamental frequency (F0) at which the call began, frequency of F0 at which the call ended, maximum frequency attained by F0, minimum frequency attained by F0, maximum frequency location (percentage of total duration), minimum frequency location (percentage of total duration). Frequency modulation (FM) sounds were defined if the maximum frequency was >20% of the minimum frequency. Noise burst (NB) sounds were defined if the sounds showed no clear F0 and no clear harmonic structure in spectral components (the "NB" does not indicate Gaussian noise in the sound signal). For classifying call types, we used criteria established by Kobayasi [18]. Briefly, calls were classified manually as being either simple syllables or composites. A simple syllable consisted of a single predominant sound element, such as an FM segment, a constant frequency (CF) segment, or an NB segment. We used a prefix to define secondary features of a call's spectrogram, e.g., AFM for arched FM and UFM for upward FM. We also used a postfix to define the duration of syllables: s = duration <75 ms and l >300 ms. A composite syllable consisted of two or more types of distinct components each representing a simple CF, NB, or FM segment combined without an interval; for example, Quasi CF-NB (QCF-NB) would indicate a QCF segment combined with an NB segment. Adding to these, we defined 'U-shape call', which is similar to UFMs but starts with downward FM. Thus, we identified nine call types in this paper (Fig 1); U-shape, UFM-s, UFM, AFM, DFM-l, DFM, QCF, NB, and QCF-NB.

U shape: syllables with downsweep followed by upsweep frequency change

UFM-s: syllables with upsweep frequency change and short duration (<75 ms)

UFM: syllables with upsweep frequency change

AFM: syllables with upsweep followed by downsweep frequency change

DFM-l: syllables with downsweep frequency change and long duration (>300 ms)

DFM: syllables with downsweep frequency change

QCF: syllables with the maximum frequency were <20% of the minimum frequency.

NB: syllables with no clear F0 and no clear harmonic structure in spectral components

QCF-NB: syllables with QCF followed by NB

After calls were recorded, we categorized calls into two groups, the high-frequency and the low-frequency vocalizations. The maximum fundamental frequency was used to categorize. We calculated the fitting functions of two normal distributions for high- and low- vocalization distributions. The intersection of the two functions is used as the objective frequency border

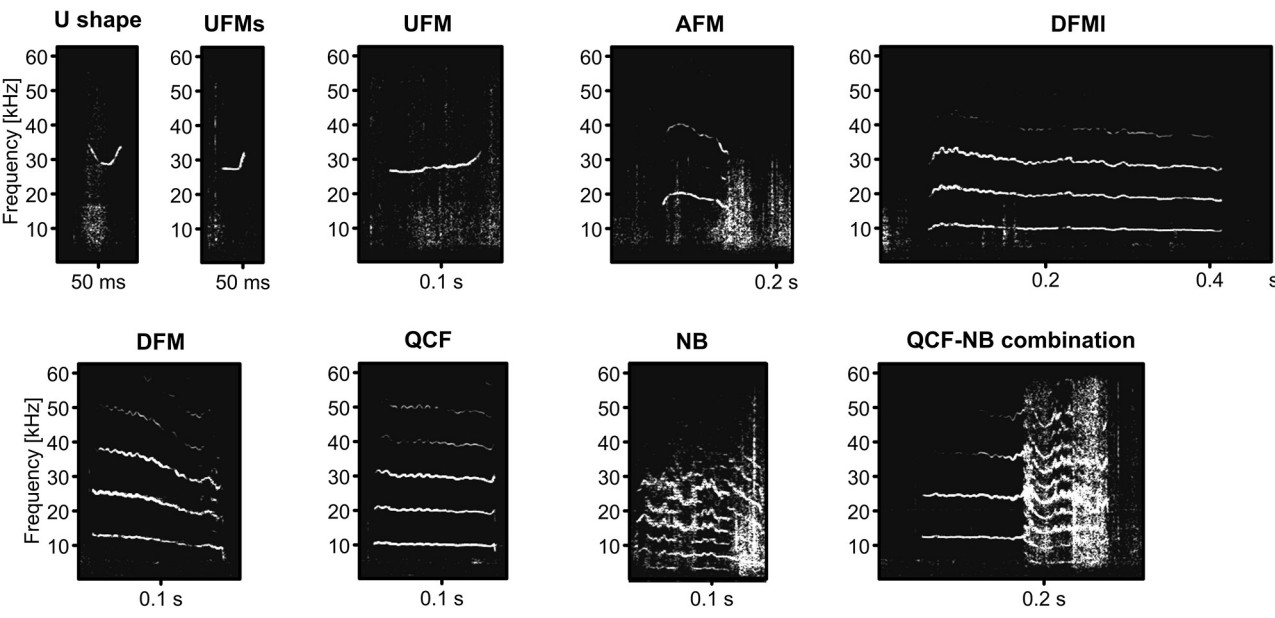

**Fig 1. Representative spectrograms of nine call types of vocalizations.**

between the high-frequency and the low-frequency vocalizations. The timing of behaviors and vocalization was synchronized with the sound in the video and the audio file.

## Statistical analysis

The Chi-square test ($\chi^2$) was used to compare the ratio of behaviors at the timing of vocalizations. To compare the medians of latency of aggressive behavior and normalized duration of aggressive behavior, we applied the Mann-Whitney U test using SPSS (IBM, Armonk, NY, US). Spearman's rank-order correlation coefficient was calculated using SPSS for each measurement combination: latency of aggressive behavior and normalized duration of aggressive behavior; latency of aggressive behavior and the number of high-frequency vocalizations per minute (the number of calls divided by the whole test duration (min)); latency of aggressive behavior and the number of low-frequency vocalizations per minute (the number of calls divided by the whole test duration (min)); normalized duration of aggressive behavior and the number of high-frequency vocalizations per minute; and normalized duration of aggressive behavior and the number of low-frequency vocalizations per minute. The Kruskal-Wallis test, followed by Dunn's test, was used for multiple comparisons. Non-paired t-test was used for comparing the mean of each call parameter before and after the first agonistic behavior in the test. Statistical significance was determined using a threshold of $p < 0.05$. Ranges enclosed within [] indicate the interquartile ranges. The standard deviations are indicated as '± numbers'.

## Results

### Same-sex resident-intruder test

We tested the effect of sexual experience on aggressive behavior in the gerbils during the same-sex resident-intruder tests. 71.4% percent (5 of 7 pairs) of VM residents and 100% (5 of 5 pairs) of EM residents exhibited aggressive behavior. The median latencies of aggressive

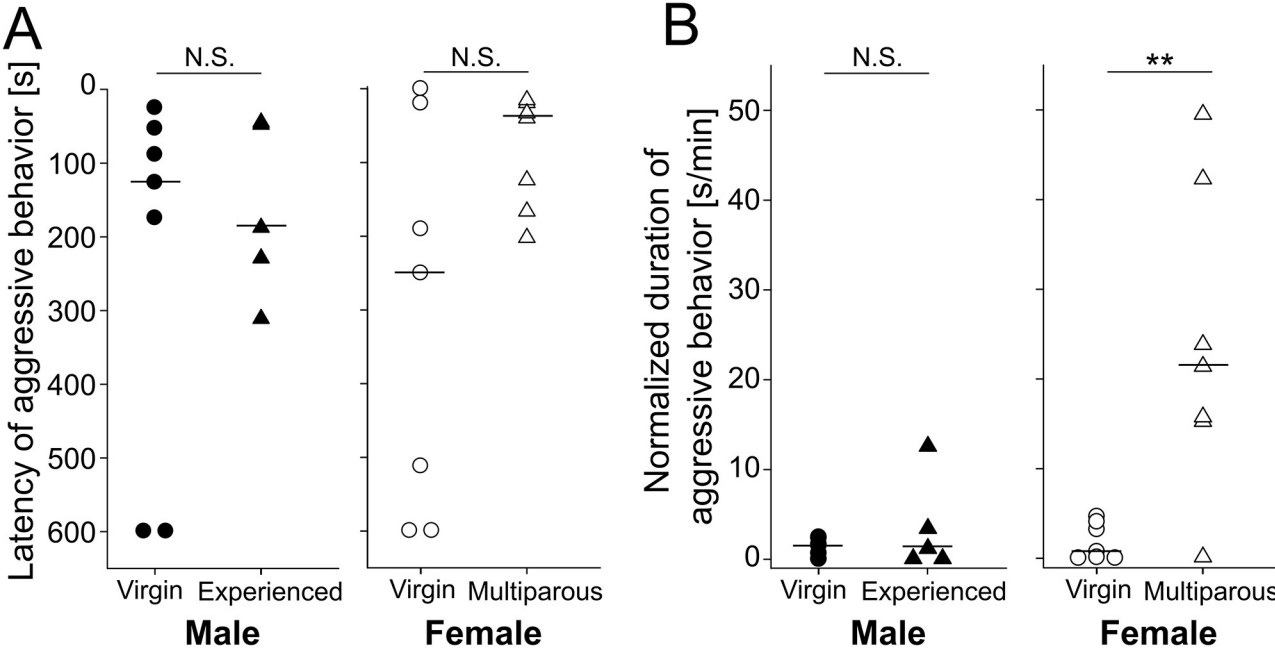

**Fig 2. Latency of aggressive behavior and normalized duration of aggressive behavior of gerbil groups during resident-intruder tests.** (A) latency of aggressive behavior in VM-VM, EM-VM, VF-VF, and MF-VF pairs. Thick horizontal bars represent medians. (B) normalized duration of aggressive behavior in VM-VM, EM-VM, VF-VF, and MF-VF pairs. Thick horizontal bars represent medians. (●, virgin male; ▲, experienced male; ●, virgin female; ▲, multiparous female). N.S.: Not significant, ** represents p < 0.01.

behavior were not significantly different between these two groups of resident males (Fig 2A *male*; VM: 127 [interquartile range: 71–388] s, EM: 188 [48–230] s, Mann-Whitney *U* test: $U = 17.0$, $p = 1$). Also, the median normalized durations of aggressive behavior were not significantly different between these two groups of resident males (Fig 2B *male*; VM: 1.5 [0.3–2.0] s, EM: 1.2 [0.1–3.4] s, $U = 21.0$, $p = 0.639$). Female pairs also showed aggressive interactions; 71.4% (5 of 7 pairs; VF resident) and 100% (7 of 7 pairs; MF resident). The median latencies of aggressive behavior were not significantly different between these two groups of resident females (Fig 2A *female*; VF: 252 [107–556] s, MF: 43 [29–147] s, $U = 13.0$, $p = 0.165$), while the median normalized duration of aggressive behavior of MF resident was significantly longer than that of VF residents (Fig 2B *female*; VF: 0.7 [0.1–3.6] s, MF: 21.4 [15.5–33.1] s, $U = 45.0$, $p = 0.007$). These results suggest that females, but not males, become more aggressive after sexual experience.

In addition, we examined the correlation between the latency of aggressive behavior and the duration of aggressive behavior. The duration of aggressive behavior in all groups decreased gradually with an increase in the latency of aggressive behavior. This correlation was statistically significant (Fig 3; Spearman's correlation coefficient, $r_s = -0.669$, n = 26, $p < 0.001$). There was no statistically significant correlation within each pair-tests.

## Vocalizations of gerbils during the same-sex resident-intruder tests

We quantified 2413 vocalizations in male-male interactions and 1949 vocalizations in female-female interactions. We detected nine call types of vocalizations (Figs 1 and 4A; U-shape, UFM-s, UFM, AFM, DFM-l, DFM, QCF, NB, and QCF-NB). Spectro-temporal features of these calls are shown in Table 2. The contour shapes of calls (Fig 1; except the U-shape call, which was newly defined in this study) in all pairs during the resident-intruder test were

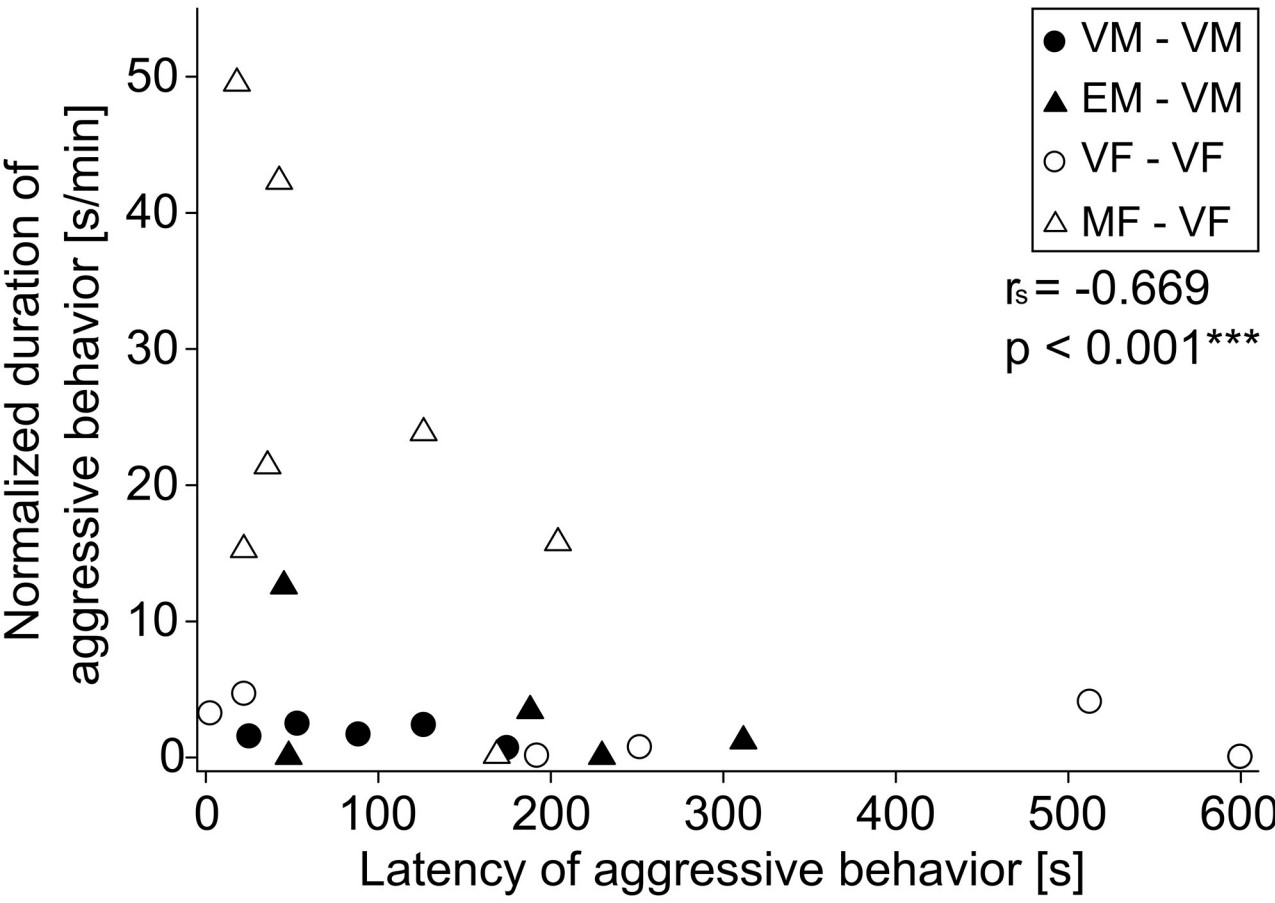

**Fig 3. Relation between latencies of aggressive behavior and durations of aggressive behavior.** Latencies of aggressive behavior and durations of aggressive behavior were negatively correlated (●, virgin male; ▲, experienced male; ●, virgin female; ▲, multiparous female; $r_s$ = -0.669; p < 0.001).

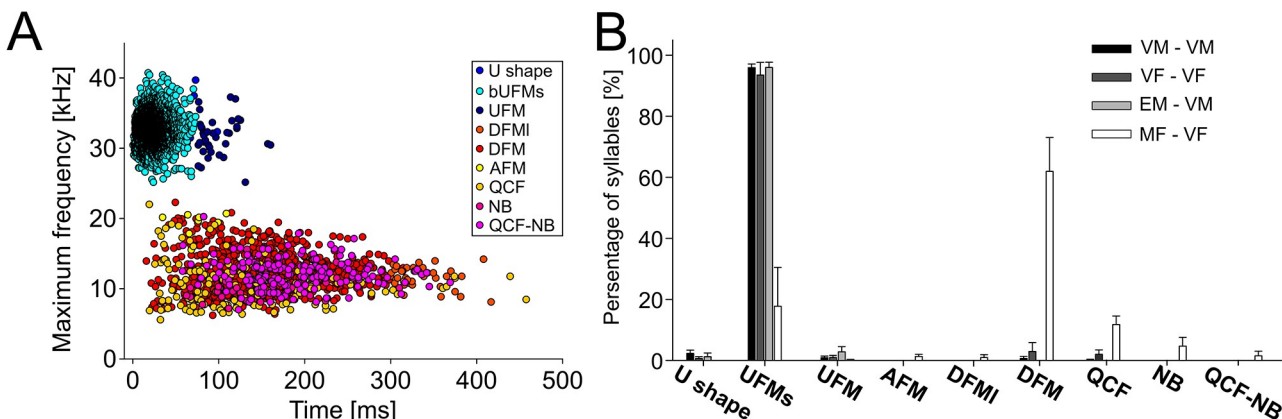

**Fig 4. Vocalizations of gerbils during resident-intruder tests.** (A) Maximum fundamental frequency and duration of each vocalization. High-frequency groups are represented by bluish symbols and low-frequency groups are represented by reddish symbols. (B) Percentages of each call type emitted during the resident-intruder test are shown.

**Table 2. Spectro-temporal features of nine call types.**

| | Initial freq. [kHz] | Terminal freq. [kHz] | Max. freq. [kHz] | Min. freq. [kHz] | Duration [ms] | Max. freq. location [%] | Min. freq. location [%] |
|---|---|---|---|---|---|---|---|
| U-shape n = 70 | 30.91 ±2.08 | 32.80 ±1.97 | 33.27 ±1.82 | 26.82 ±4.70 | 39.30 ±12.23 | 78.32 ±40.37 | 54.99 ±21.71 |
| UFM-s n = 3237 | 28.04 ±1.83 | 32.17 ±2.29 | 32.51 ±2.17 | 26.77 ±5.02 | 25.03 ±13.20 | 92.43 ±21.69 | 24.46 ±27.54 |
| UFM n = 22 | 26.39 ±3.84 | 29.90 ±4.85 | 30.36 ±6.00 | 21.34 ±9.00 | 94.82 ±20.67 | 74.60 ±37.71 | 57.77 ±36.34 |
| AFM n = 8 | 16.42 ±3.12 | 14.17 ±2.44 | 17.79 ±3.53 | 12.09 ±5.11 | 78.03 ±27.65 | 24.05 ±19.32 | 97.41 ±5.63 |
| DFM-l n = 44 | 10.83 ±1.22 | 7.87 ±1.37 | 11.62 ±1.24 | 5.04 ±3.52 | 337.19 ±30.21 | 7.68 ±12.91 | 90.90 ±20.37 |
| DFM n = 700 | 12.12 ±2.85 | 7.36 ±2.55 | 12.80 ±3.10 | 6.55 ±2.78 | 149.23 ±58.83 | 7.58 ±15.38 | 90.76 ±19.03 |
| QCF n = 157 | 10.74 ±3.59 | 9.81 ±3.70 | 12.04 ±3.98 | 7.88 ±4.54 | 126.83 ±76.83 | 16.95 ±22.78 | 82.96 ±29.63 |
| NB n = 26 | 9.70 ±2.23 | 7.11 ±1.66 | 11.22 ±2.43 | 4.79 ±2.16 | 155.97 ±77.44 | 15.83 ±22.75 | 73.09 ±30.42 |
| QCF-NB n = 98 | 11.66 ±1.54 | 7.82 ±1.89 | 12.87 ±1.74 | 5.41 ±2.54 | 199.62 ±70.10 | 9.11 ±13.94 | 86.42 ±21.22 |

Spectro-temporal features of nine call types are shown. Frequency of fundamental frequency (F0) at which the call began (Initial freq.), frequency of F0 at which the call ended (Terminal freq.), maximum frequency attained by F0 (Max. freq.), minimum frequency attained by F0 (Min. freq.), total duration (Duration), maximum frequency location (percentage of total duration; Max. freq. location), and minimum frequency location (percentage of total duration; Min. freq. location) are shown. Mean ±S.D. are indicated.

similar to those reported in previous studies [18]. The percentages of each vocalization among the total calls during the tests are shown in Fig 4B. Higher percentages of DFM/QCF/NB and a lower percentage of UFM-s were observed in MF (resident)—VF (intruder) pairs than in the other pairs. Obviously, as shown in Fig 4A, the nine call types could be categorized into two groups. From the fitting functions of two normal distributions, the separation at 24.6 kHz was determined. Thus, for further analysis, we categorized the quantified nine types of vocalizations into two groups: high-frequency with short duration (U-shape, UFM-s, and UFM; 32.7 ± 2.0 kHz, 25.9 ± 14.7 ms; n = 3329), and low-frequency with long duration (AFM, DFM-l, DFM, QCF, NB, and QCF-NB; 12.1 ± 2.9 kHz, 160.4 ± 73.3 ms; n = 1033).

We compared the number of calls per minute during the tests between the four tests. The median numbers of high-frequency calls per minute ranged from: 20.9 [12.5–24.5] in VM-VM, 4.0 [3.5–13.2] in EM-VM, 8.0 [6.1–10.5] in VF-VF, and 0.1 [0.0–3.1] in MF-VF (Fig 5A). The median numbers of low-frequency calls per minute ranged from: 0.0 [0.0–0.1] in VM-VM, 0.0 [0.0–0.0] in EM-VM, 0.1 [0.0–0.2] in VF-VF, and 13.6 [7.4–25.4] in MF-VF (Fig 5B). Vocalizations of MF-VF pairs exhibited a dramatically different distribution compared to those of the other groups. The rate of low-frequency calls was significantly higher in the MF-VF interaction than in the other pairs (Fig 5B; MF-VF vs. VM-VM, p < 0.05; MF-VF vs. EM-VM, p < 0.001; MF-VF vs. VF-VF, p < 0.05; Kruskal-Wallis test followed by Dunn test). For high-frequency vocalizations, VM-VM pairs emitted calls more frequently than MF-VF pairs (Fig 5A; MF-VF vs. VM-VM, p = 0.04; Kruskal-Wallis test followed by Dunn test). These results suggest that sexual dimorphism and sexual experience affected not only behavior but also vocalizations associated with aggressiveness. In addition, there was a negative correlation between the numbers of high- and low-frequency calls (Fig 5C; $r_s$ = -0.413, p = 0.036; n = 26). There was no statistically significant correlation within each pair-tests.

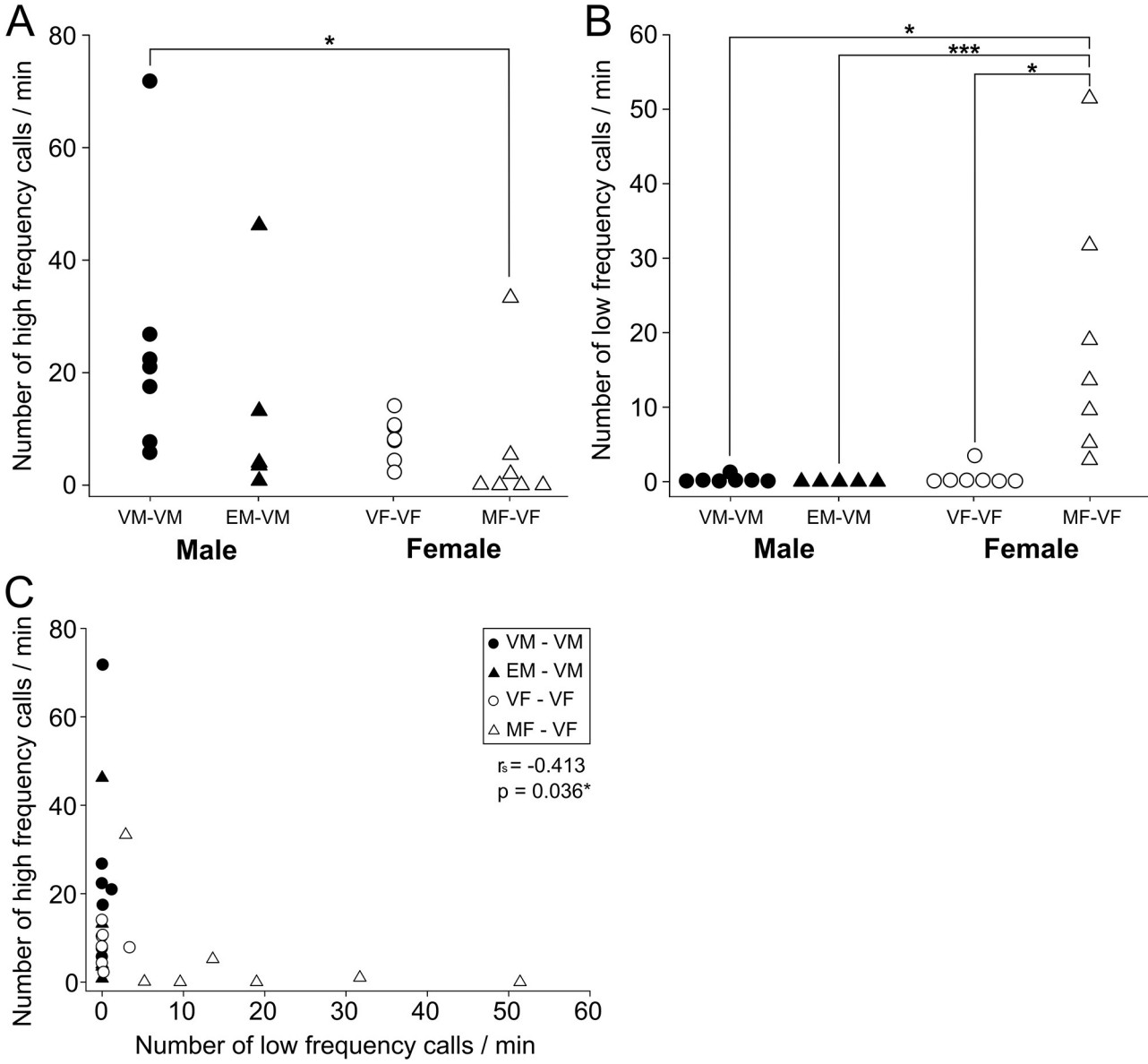

**Fig 5. Numbers of two call types by during resident-intruder pairs.** (A) Numbers of high-frequency calls per minute by pair-tests. VM-VM pairs emitted high-frequency calls much more than MF-VF pairs. (B) Numbers of low-frequency calls per minute by pair-tests. MF-VF pairs emitted low-frequency calls much more than the other pairs. (C) There is a negative correlation between the numbers of high-frequency calls and that of low-frequency calls ($r_s$ = -0.413; p = 0.036). Each symbol represents calls from a single test (●, virgin male; ▲, experienced male; ●, virgin female; ▲, multiparous female). * represents p < 0.05. *** represents p < 0.001.

## Relationship between each vocalization and aggressive interactions during the same-sex resident-intruder tests

First, we created the ethograms for 26 tests (examples shown in Fig 6A) and categorized the behaviors into two categories, non-agonistic or agonistic (indicated by blue or red in Fig 6A). Then, we quantified the type of behavior at the timing of high- or low-frequency calls emitted (Fig 6B). As shown in Fig 6B, high-frequency calls mostly coincided with no conflicts between the resident and intruder (96.2%), while aggressive interactions were associated much more frequently with low-frequency calls than with high-frequency calls (low, 45.0%; high, 3.8%;

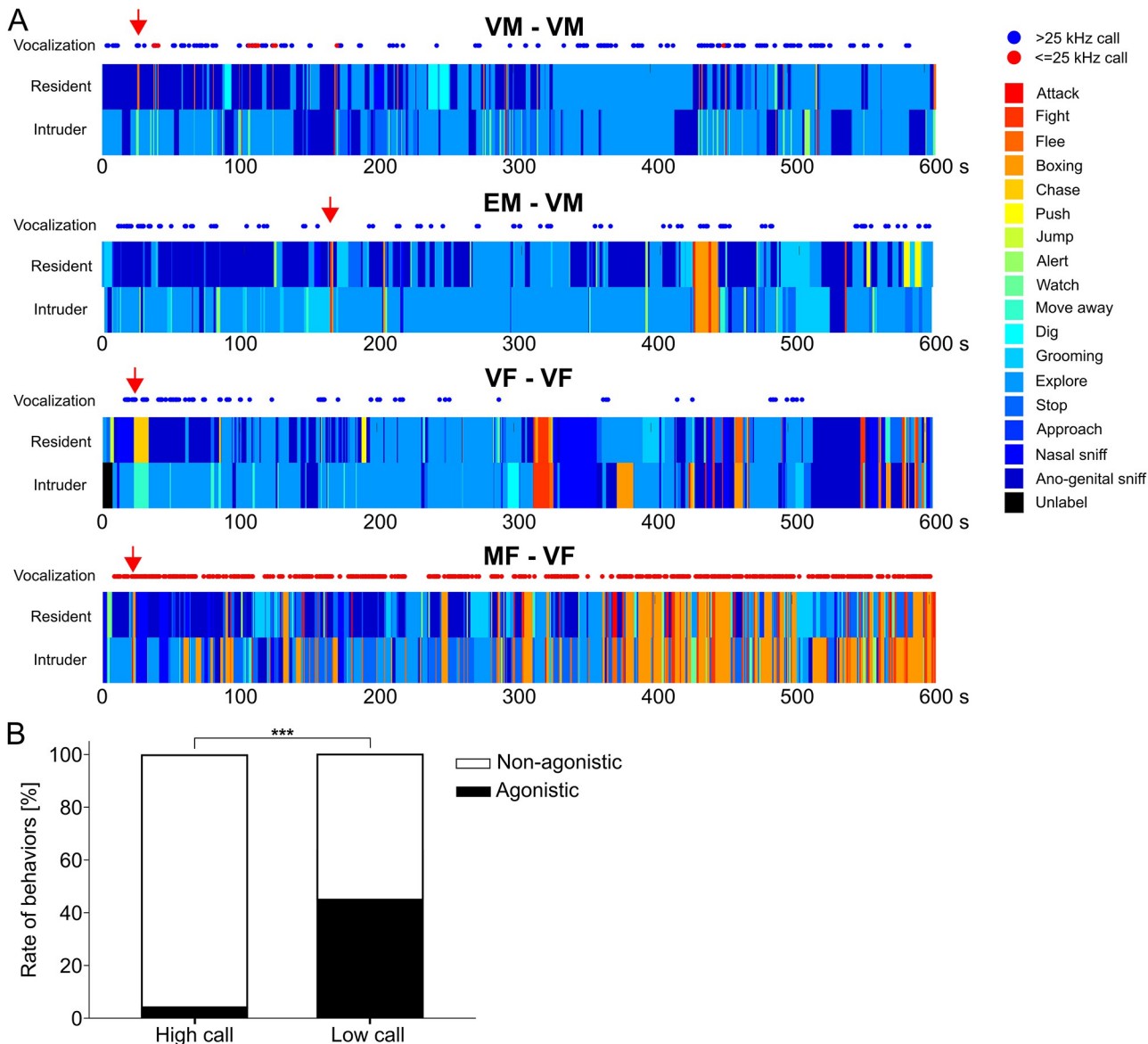

**Fig 6. Relation between two types of calls and each behavior.** (A) Representative ethograms of each pair-test during resident-intruder tests. Ethograms for VM-VM, EM-VM, VF-VF, and MF-VF pairs were aligned from top to bottom. Bluish colors represent non-agonistic behaviors and reddish colors represent agonistic behaviors. Blue circles and red circles indicate >24.6 kHz calls and 24.6≤ kHz calls respectively. Red arrows indicate the timing when the resident initiated the first aggressive behavior. MF-VF pairs exhibited different patterns of vocalizations and behaviors. (B) The ratio of behaviors at the timing each call emitted. At the timing of high-frequency calls, agonistic interactions were rarely observed, contrary, at the timing of low-frequency calls, agonistic interactions were dominant. *** represents p < 0.001.

Pearson's $\chi^2$ = 1138.5, p < 0.001). These results suggest that high-frequency calls accompany non-agonistic interactions, whereas low-frequency calls are related to agonistic interactions. Next, we analyzed the correlation between aggressive behaviors and numbers of the two types of vocalizations. Normalized durations of aggressive behavior correlated negatively with the number of high-frequency vocalizations (Fig 7A and 7C; latency, $r_s$ = 0.165, p = 0.421; duration, $r_s$ = -0.479, p = 0.013; n = 26). In contrast, the latency of aggressive behavior and duration of aggressive behavior correlated negatively and positively, respectively, with the number of low-frequency vocalizations (Fig 7B and 7D; latency, $r_s$ = -0.419, p = 0.033; duration, $r_s$ =

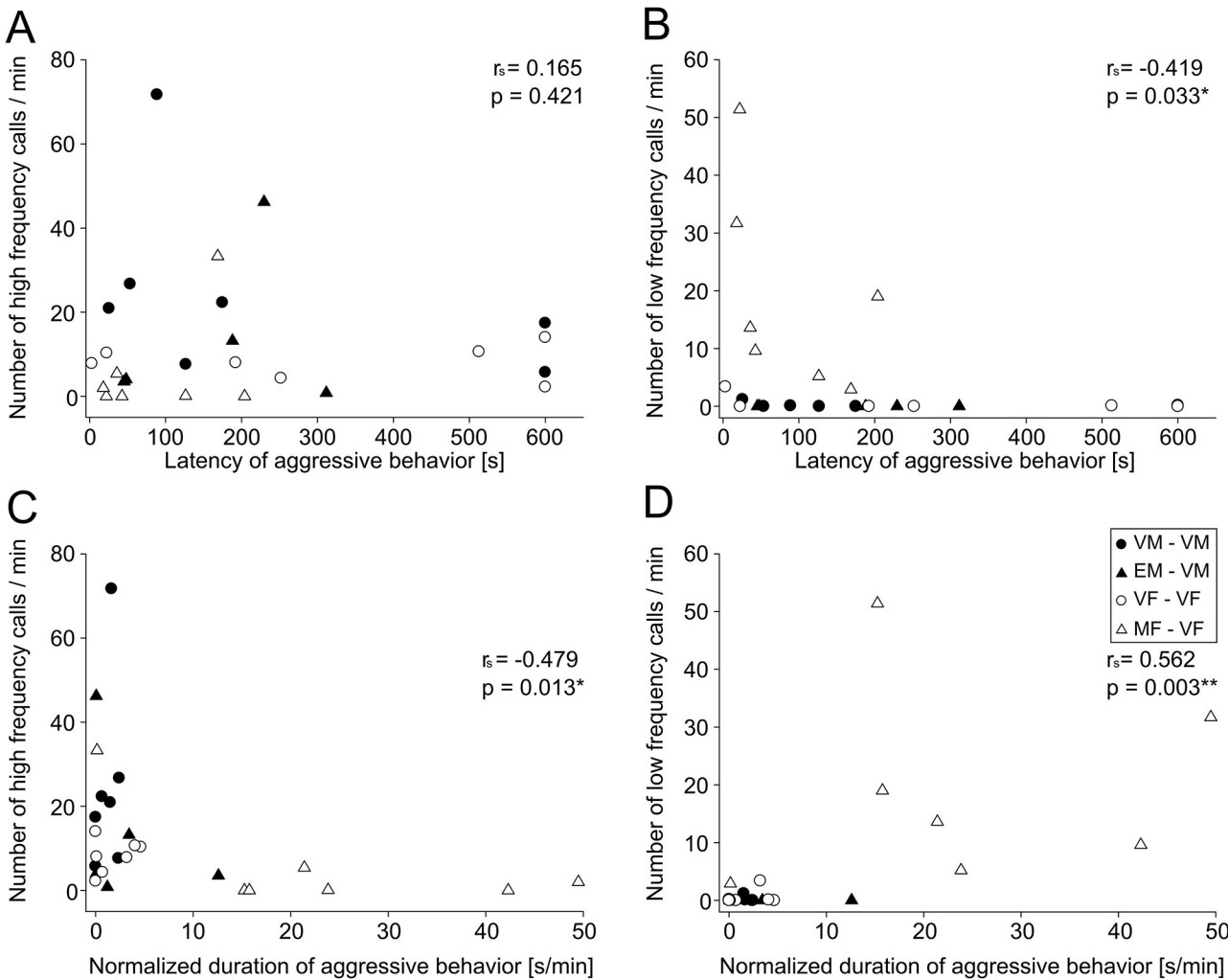

**Fig 7. Relationship between two types of calls and aggressive behaviors.** (A) Distributions of latency of aggressive behavior and rate of high-frequency vocalizations. (B) Distributions of latency of aggressive behavior and rate of low-frequency vocalizations. (C) Distributions of duration of aggressive behavior and rate of high-frequency vocalizations. (D) Distributions of duration of aggressive behavior and rate of low-frequency vocalizations. Each symbol represents vocalizations from a single test (●, virgin male; ▲, experienced male; ●, virgin female; ▲, multiparous female). * represents p < 0.05. ** represents p < 0.01.

0.562, p = 0.003; n = 26). There was no statistically significant correlation within each pair-tests. These results also support the idea that high-frequency calls accompany non-agonistic interactions while low-frequency calls are related to agonistic interactions. Finally, we compared five parameters (Initial freq., Terminal freq., Max. freq., Min. freq., and Duration; refer to Table 2) of high- and low-frequency vocalizations before and after the first agonistic behavior in the test. After the first agonis tic event, Terminal freq. and Max. freq. of high-frequency vocalizations shifted to slightly high frequency (Terminal freq., form 32.28 ± 2.13 kHz to 32.63 ± 2.27 kHz, p < 0.001; Max. freq., form 32.43 ± 2.08 kHz to 32.85 ± 2.07 kHz, p < 0.001, n (before) = 1167, n (after) = 2162). Initial freq., Max. freq., and Min. freq. of low-frequency vocalizations also shifted around 1 kHz higher frequency (Initial freq., form 10.35 ± 2.47 kHz to 11.96 ± 2.91 kHz, p = 0.004; Max. freq., form 10.82 ± 3.67 kHz to 12.37 ± 3.22 kHz, p = 0.013; Min. freq., form 6.44 ± 3.32 kHz to 7.9 ± 2.72 kHz, p = 0.006, n (before) = 28, n

(after) = 1005). The duration of low-frequency vocalizations shifted significantly longer after the first agonistic event (Duration, form 105.39 ± 59.7 ms to 162.82 ± 75.9 ms, p < 0.001).

## Discussion

We showed that both sexes of gerbils, especially MFs, exhibited aggressive responses toward intruders. High-frequency calls were related with non-agonistic interactions, while low-frequency calls, which might represent aggression, were more related to agonistic interactions than non-agonistic interactions. Low-frequency calls were mostly observed in MF–VF pairs, the pair-test that showed the most aggressive interaction. This is the first report that quantified the relationship between each call and aggressive behaviors and demonstrated the modulations of vocalization during agonistic encounters by sexual experiences in Mongolian gerbils.

We confirmed that male gerbils showed aggressive behaviors, and neither the latency of aggressive behavior nor the duration of aggressive behavior was significantly different between VM and EM (Fig 2). Many previous reports have studied aggressive interactions in gerbils. A study [27] showed that gerbils attack unfamiliar conspecifics and other rodent species. Moreover, EMs tend to be more aggressive toward VFs [13]. VMs exhibit aggressive behavior toward same-sex intruders [16]. On the other hand, another study reported that aggressive behaviors were not observed in VM-VM interactions [28]. In addition, another study showed that males were more aggressive during parental care of pups compared with the period of cohabitation with females [29]. Divergence among reports may be attributed to differences in the testing environment or breeding conditions. Mongolian gerbils exhibit territorial aggression in a considerably large territory [11,12]. Therefore, the apparatus size for the resident-intruder test might have effects on aggressiveness. In larger areas (e.g., 3m × 3m field), resident gerbils might exhibit a different repertoire of aggressive behaviors toward the intruder. This factor should be carefully examined in future studies. Furthermore, we defined males who experienced both mating and parenting as sexually experienced. If parenting is not a part of the criteria in other studies, this might cause differences between the previous and present findings.

We also observed aggressive behaviors in females, especially in MFs. Our data indicate that aggression in female gerbils was increased by parenting experience. Indeed, the normalized duration of aggressive behavior of the MFs was longer than that of the VFs (Fig 1). Aggressive behaviors in both virgin and multiparous female gerbils have been reported in previous studies. VF Mongolian gerbils exhibit aggressive behavior during same-sex encounters [15]. MF gerbils attack VMs more frequently [13] and VFs [17] than VFs do. Our results are consistent with these previous findings. It is well known that females of laboratory rodents, such as rats and mice, that are not lactating or pregnant are less aggressive than males [3,30–32]. From this point of view, MF gerbils could be a suitable model for studying female aggression.

Several studies in rodents have debated the relationship between aggressive behavior and the estrus cycle in some rodents. Ovariectomy reduces aggression in bank voles (*Myodes glareolus*) [33] and Syrian hamsters (*Mesocricetus auratus*) [34]. On the other hand, a study showed that female Wistar rats attacked an intruder female, independent of their estrous cycle [35]. Similarly, aggression was not affected by the estrous stage in female Siberian hamsters (*Phodopus sungorus*) [36]. A recent study showed that diestrus female Mongolian gerbils exhibited aggressive behaviors, but the study did not compare diestrus females with estrus females [15]. In the present study, the estrus cycle was not determined in any of the groups of female gerbils. Further studies are needed to investigate the relationship between aggressive behavior and the estrus cycle in Mongolian gerbils.

We found that gerbils emitted nine types of vocalizations during the resident-intruder test. The spectral shapes of these calls were basically consistent with those reported in previous

studies [17,18]. Alert calls described in the same study [17] were not observed in our resident-intruder tests. The nine call types are categorized into two groups; high-frequency vocalization and low-frequency vocalization. It has been reported that Mongolian gerbils emit high-frequency vocalizations during amicable interactions, while low-frequency vocalizations seem to relate to aggression [17,18]. Mongolian gerbils have the good ability to perceive low-frequency sounds than the other small rodents, such as rats and mice [37]. Low-frequency sounds could conduct for a long distance. Hence, in the natural environment, Mongolian gerbils might use the low-frequency vocalizations to communicate with distant partners rather than the high-frequency vocalizations. Several studies in other rodents also have investigated the differences in vocalization frequency in different contexts. Some reports confirmed that rats emit 50 kHz high-frequency vocalizations when they approach, investigate their partners [4,38], and even in the aggressive situation as well as play [39]. Similarly, the number of ultrasonic vocalizations (USVs) among male mice increases with an increase in interaction time [40], and resident female mice emit USVs when interacting with a novel female intruder [41]. Broadband calls, which are composed with multiple harmonics at the range of 1–20 kHz, of Siberian hamsters were produced during aggressive behavior [42]. The bark calls, which have a fundamental frequency around 20 kHz with multiple harmonics, were positively correlated with an increase in aggressive behavior in California mice (*Peromyscus acalifornicus*) [43]. Some previous studies reported that the 22 kHz vocalizations emitted by an intruder inhibited biting attacks from the resident rat [4,44,45], while another report did not confirm this effect [46]. Morton has claimed an idea that mammals and birds use the low-frequency sounds when hostile and use higher frequency when frightened, appeasing, or approaching in a friendly manner, by reviewing a number of literatures [47]. Collectively, previous reports suggested that higher frequency calls accompanied friendly or non-agonistic interactions, while lower frequency calls related to agonistic conflicts in many rodents.

Consistent with this idea, our results showed that high-frequency vocalizations were accompanied by a reduction of the duration of aggressive behavior. This implies that vocalizations might have some influence on behavioral output including aggressiveness. However, it remains unknown whether this high-frequency call develops an amicable relationship with the intruder or merely represents lower aggression of both the residents and intruders. Here, we observed that high-frequency vocalization in gerbils was strongly related to lower conflicts between the two individuals. Some reports studied the influence of 50 kHz USV on inter-individual behaviors in rats. Devocalization caused the reduction of play behaviors, while the aggressive behaviors were increased [39,48]. Also, the playback experiment of 50 kHz USV promoted social approach behaviors [49]. These previous findings suggest that high-frequency vocalizations affect on behavioral interactions and help developing an amicable relationship at least in rat.

As for low-frequency vocalizations, the numbers of low-frequency calls had a positive correlation with agonistic interactions during the resident-intruder test and many low-frequency calls were observed during not only agonistic interactions but also during the entire period of the resident-intruder test. Behavioral repertories during agonistic interactions consist of multiple subsets, including physical attacks, vocalizations, and changing posture. Indeed, orchestrated multiple responses, such as combinations of biting, arching, and hissing, are known to be triggered by electrical stimulation of the hypothalamus in cats [50,51]. These responses (except for physical attack) are less apparent and have not been studied as much in other laboratory rodents. Low-frequency calls emitted in MF-VF gerbil interactions would be a good characteristic behavior to study multiple aspects of the behavioral response during agonistic events in rodents. In addition, there is a possibility that low-frequency vocalizations could be linked to not only agonistic interactions but also important characteristics of a vocalizer, such

as sex, reproductive state, or social experiences. Further studies are required to examine these possibilities.

Finally, we showed that MF-VF pairs emitted low-frequency vocalizations more, while high-frequency vocalizations less, compared with that in the other three groups (Fig 4B). Even though our experiment could not determine whether the resident or intruder emitted a specific call, probably MFs (resident) would emit low-frequency calls during the resident-intruder test since low-frequency calls were rarely observed in VF-VF interactions. Of course, as reported by Ter-Mikaelian et al. [17], a submissive subject could also emit these low-frequency calls. We assume that this actually happened to some extent. However, given that low-frequency calls were recorded even during the period before the resident started attacking the intruder in MF-VF pairs (this was not observed in the other groups), it could be possible that some of low-frequency calls were emitted by MF residents.

These results indicated that vocalizations induced in a specific context could also be modulated by past experiences concerning reproduction. A previous study showed that male mice emitted more USVs after the sociosexual experience [52]. Low-frequency vocalizations increased during same-sex encounters in female pair-bonded California mice [43]. Collectively, these reports and our findings indicate that vocalizations could be modulated by mating experience. However, it remains unknown why MF-VF pair emitted more low-frequency calls compared with those in the other three pairs. We hypothesize that vocalizations are influenced by hormonal changes induced by parenting experiences in female Mongolian gerbils. To elucidate how hormonal changes trigger different patterns of vocalization and the main role of this change, we will carry out refined manipulations of emotion- and vocalization-related neural systems in the future.

## Supporting information

**S1 Dataset.**
(XLSX)

## Acknowledgments

We wish to thank S. Muramoto, S. Tsukurimichi, and K. Hori for their assistance.

## Author Contributions

**Conceptualization:** Ryo Yamamoto.

**Data curation:** Takafumi Furuyama, Munenori Ono, Nobuo Kato, Ryo Yamamoto.

**Formal analysis:** Takafumi Furuyama, Munenori Ono, Sachiko Yamaki.

**Funding acquisition:** Takafumi Furuyama, Munenori Ono, Sachiko Yamaki, Nobuo Kato, Ryo Yamamoto.

**Investigation:** Takafumi Furuyama, Takafumi Shigeyama, Ryo Yamamoto.

**Methodology:** Takafumi Furuyama, Ryo Yamamoto.

**Project administration:** Takafumi Furuyama, Ryo Yamamoto.

**Software:** Takafumi Furuyama.

**Supervision:** Takafumi Furuyama, Kohta I. Kobayasi, Nobuo Kato, Ryo Yamamoto.

**Validation:** Takafumi Furuyama, Ryo Yamamoto.

**Visualization:** Takafumi Furuyama.

**Writing – original draft:** Takafumi Furuyama, Ryo Yamamoto.

**Writing – review & editing:** Takafumi Furuyama, Munenori Ono, Kohta I. Kobayasi, Nobuo Kato, Ryo Yamamoto.

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
