## [Decision Letter · Decision Letter 0]

22 Feb 2022

PONE-D-21-38093Vocalization during agonistic encounter in Mongolian gerbilsPLOS ONE

Dear Dr. Yamamoto,

Thank you for submitting your manuscript to PLOS ONE. After careful consideration, we feel that it has merit but does not fully meet PLOS ONE’s publication criteria as it currently stands. Therefore, we invite you to submit a revised version of the manuscript that addresses the points raised during the review process.

As you see, all reviewers are positive about your study. Nevertheless, they have concerns with the present manuscript.

It would be best if you clear the aim of the study up. It would help to formulate a hypothesis for each task and reorganize the text accordingly. You could use subheadings.

In the Methods, follow Reviewer 1’s suggestions regarding determining who is who in the encounter and how you categorized the spectrograms. Will you complete citations for individual elements of agonistic behaviour? Is there any reason you used virgins for the intruders’ test? Could you consider the structure of vocalizations before and after the attack?

Will you be more specific when saying “for at least one week“, “many of the residents”, “the vast majority”, etc.? It seems to me you used the subjects repeatedly on the tests. If not, you should clearly say it. If so, however, what was the order of the type of tests? How did you measure the number of high-frequency vocalizations per minute and the number of low-frequency vocalizations per minute?

Please consider multifactorial tests rather than non-parametric simple test statistics. Your analysis did not show the interrelationships between the factors, such as the age of the subjects, their sexual experience, the number of animals kept in one plexiglass cage, the latency and duration of aggressive behaviours, etc. The uncategorized correlation in Fig. 2 is misleading. You present a negative correlation. However, it seems the case for MF-VF, not that much for EM-VM, and it is unlikely for the rest of the categories. (Similarly, in Fig. 6 B, C, and D.) A multifactorial approach would help.

It would also help if you strictly separated the Methods, Results and Discussion throughout the text. Scientific names in the first mentioning of the species are missing. You should transfer part of the Figure captions into the main text where appropriate. Captions should not contain the results at all. Be so kind and also consider the Reviewers’ detailed suggestions.

Please use the line numbers throughout the text if you decide to submit the revision. Not the numbering for each page. It might also help to understand better if you used conventional division to ultrasonic calls (over 20 kHz) rather than using your division (>25 kHz).

We look forward to receiving your revised manuscript.

Kind regards,

Ludek Bartos

Academic Editor

PLOS ONE

Journal Requirements:

Reviewers' comments:

Reviewer's Responses to Questions

**Comments to the Author**

1. Is the manuscript technically sound, and do the data support the conclusions?

Reviewer #1: No

Reviewer #2: Yes

Reviewer #3: Yes

2. Has the statistical analysis been performed appropriately and rigorously? 

Reviewer #1: I Don't Know

Reviewer #2: Yes

Reviewer #3: Yes

3. Have the authors made all data underlying the findings in their manuscript fully available?

Reviewer #1: No

Reviewer #2: Yes

Reviewer #3: Yes

4. Is the manuscript presented in an intelligible fashion and written in standard English?

Reviewer #1: No

Reviewer #2: No

Reviewer #3: Yes

5. Review Comments to the Author

Reviewer #1: This MS is focused on relationship between agonistic behaviour and the number of high-frequency (above 25 kHz) and low-frequency (below 25 kHz) vocalizations during same-sex dyad encounters of sexually experienced and unexperienced (virgin) male and female Mongolian gerbils. The call rate of high-frequency vocalizations was negatively correlated with the duration of aggressive behavior, while the call rate of low-frequency vocalizations was negatively correlated with the latency of aggressive behavior and positively correlated with the duration of aggressive behavior.

I found the following main problems with MS.

The aim of this study not well formulated and should be re-written.

Introduction is focused on the Mongolian gerbils and does not provide the understanding the place of this particular study within the investigated problem, although many similar studies have been conducted on rats, mice and other rodents, including the gerbils.

In Methods, the authors do not indicate, how it was determined who is the caller during aggression, the resident (attacker) or intruder (defender). These limitations should be clearly explained. The authors should indicate, on what they based their separation to the aggressive/agonistic and non-agonistic behaviors. The authors should indicate how they classified the calls to the high- and low-frequency. Was it done based on visual inspection of spectrograms or measuring fundamental frequency?

The Results often represent mix of results and discussion; this should be clearly separated in the revised version.

Discussion is looking as some draft of literature review rather than discussing own results, and should be re-written concisely and logically, in relation to the obtained results. Discussion also contains and discusses the conclusions which are not supported by the results of this study.

The MS is difficult to read, although English is mostly appropriate. In the revised version, the authors should provide line numbers throughout the text, because numeration of each paper separately complicates enormously the work of Reviewer, Editor and Authors themselves.

Abstract

L. 9-10 We also confirmed two types of vocalisations: high-frequency (>25 kHz) and low-frequency (<25 kHz)

This is strange subdivision. Commonly, classifying calls in the Mongolian gerbil is to ultrasonic calls over 20 kHz and human-audible calls below 20 kHz.

L. 10 with multiple harmonics

This has not sense. Harmonics are integer multiples of fundamental frequency (the lowest frequency band). If the fundamental frequency is low, they are spaced more densely, if the fundamental frequency is high, they are spaces more rarely. So, you just do not see the harmonics of high-frequency calls.

L. 10-14

The call rate of high-frequency vocalizations was negatively correlated with the duration of aggressive behavior, while the call rate of low-frequency vocalizations was negatively correlated with the latency of aggressive behavior and positively correlated with the duration of aggressive behavior.

Who vocalized during the aggressive encounters: the victim, the victor, both fighters?

Introduction

L. 14-15 As it is well known, laboratory rodents such as mice or rats rarely exhibit this extent of aggression.

This statement needs in references, please provide them.

L. 1-6. Although several studies cited in the previous paragraph have reported aggression-specific vocalizations, it remains unclear how sex and sexual experience affect the call rate of various vocalizations in Mongolian gerbils. Thus, we analyzed vocalizations between four different pairs (virgin male–virgin male; experienced male– virgin male; virgin female–virgin female; multiparous female–virgin female) of Mongolian gerbils through the same-sex resident-intruder test.

The aim of the study is poorly formulated and does not reflect the content of the paper. Did you analysed vocalizations, call rates and behaviour latencies? For what? Please re-write the aim accordingly to you real results.

Methods

Same-sex resident-intruder test

L 16 We conducted resident-intruder tests

Добавьте 26 перед resident-intruder tests

L 29 - 04 The observed behaviors were classified as follows: alert posture, ano-genital sniff, approach, attack, clinch, chase, dig, explore, fight, flee, jump, move away, nasal sniff, push, self-groom, stop moving, and watch. ‘attack’, ‘chase’, ‘clinch’, ‘flee’, and ‘push’ were scored as aggressive/agonistic behaviors. The rest was scored as non-agonistic interaction.

It is necessary to provide the references to papers (they are numerous) and/or short descriptions of the forms of aggressive and non-aggressive behaviour of rodents, to make clear for the reader, on what you based your separation to the aggressive/agonistic and non-agonistic behaviors. Why fight is not assigned to the aggressive/agonistic behaviors?

Recording and analysis of vocalizations

L. 24 Sonograms

Thereafter you use more correct term “spectrogram”. Use it here too.

L 1 Delete total duration after F0

L 16-17 Thus, we identified nine calls in this paper; U-shape, UFM-s, UFM, AFM, DFM-l, DFM, QCF, NB, and QCF-NB

Please provide here as an illustration Fig.3A as Fig.

Please make it clear, whether these nine call types could be only low-frequency or only high-frequency or both low- and high-frequency?

The authors should indicate how they classified the calls to the high- and low-frequency (for example, if F0max=30 kHz and F0min=20 kHz). Was it done based on visual inspection of spectrograms or measuring fundamental frequency?

Statistical analysis

L 8-9 number of high-frequency vocalizations per minute; latency of aggressive behavior and the number of low-frequency vocalizations per minute;

Please explain in the previous section, how did you measure the number of high-frequency vocalizations per minute and number of low-frequency vocalizations per minute. From the entire test duration or only from the duration after the first aggression?

Results

Same-sex resident-intruder test

L 3-4 We conducted resident-intruder tests to assess the extent of territorial aggression in gerbils. All intruders were virgin and of the same sex as the resident.

Delete, these are not the results

L 4-7 Many of the residents expressed aggressive behaviors toward the intruder for several minutes and frequently emitted calls. The vast majority of aggressive behaviors were initiated by residents, thus basically we focused on the behaviors of residents in this study.

Please decode what are Many of the residents and The vast majority. In the Results, you should provide the digits, not judgements.

L 4-5 Many of the residents expressed aggressive behaviors toward the intruder for several minutes and frequently emitted calls

How the authors determined who is calling, the resident or the intruder? In gerbils, both can call. The low-frequency calls produce the fleeing animals during aggressive contacts (Ter-Mikaelian et al. 2012, cited in MS). This is a key question, which should be addressed in the Methods.

Analysis of vocalizations of gerbils during the same-sex resident-intruder tests

L. 14-15. Analysis of vocalizations of gerbils during the same-sex resident-intruder tests

Delete “Analysis of”. Here, you describe the results of analysis rather than analysis itself.

L 16 Gerbils frequently emitted calls during the resident-intruder test

Delete this sentence

L. 18. We detected nine calls of vocalizations

This sounds senseless. Do you mean “we detected nine call types”?

L 20-23 The spectral shapes of calls (Fig. 3A; except the U-shape call, which was newly defined in this study) in all pairs during the resident-intruder test were similar to those reported in previous studies [17, 18]. Alert calls described in the same study [17] were not observed in our resident-intruder tests.

This belongs to Discussion. Data from previous studies should not be discussed in the Results section.

L. 20 spectral shapes

This sounds unclear and imprecise. Replace with “contour shapes”.

L. 17-18 It has been reported that the frequency of vocalizations bears semantic significance with a dividing line at 25 kHz [17, 18]. Vocalizations with a maximum fundamental frequency above 25 kHz were observed during non-agonistic interactions, whereas those below 25 kHz were observed during agonistic interactions. The separation at 25 kHz is consistent with our observations (Fig. 3B).

This part of text represents a mix of Methods, own Results and Discussion. It should be separated accordingly to these sections. Please substantiate the separation to the low-frequency and high-frequency calls in Introduction. Please provide the criteria for separation to these categories in Methods. Otherwise, Statistical analysis remains perfectly unclear.

L. 3-5 high-frequency with short duration (U-shape, UFM-s, and UFM; 32.7 ± 2.0 kHz, 25.9 ± 14.7 ms), and low-frequency with long duration (AFM, DFM-l, DFM, QCF, NB, and QCF-NB; 12.1 ± 2.9 kHz, 160.4 ± 73.3 ms)

Please indicate that the fundamental frequency (maximal?) and the duration are provided for all high-frequency and low-frequency криков. Please indicate the numbers of the high-frequency and low-frequency calls.

Fig. 3 legend

Replace “submitted to” with “in”

Fig. 3 legend MF-VF pairs seemed to have a different composition of syllables from the other pairs.

Transfer this from the figure legend to main text.

Table 1. Spectro-temporal features of nine calls.

Replace “nine calls” “nine call types” here and in the sentence under the Table 1. Please add to the legend designations/ decoding of all acoustic variables presented in the column headings.

Table 1. Spectro-temporal features of nine calls.

Please indicate n (call number) for each call type

Table 1. UFM

Mean Min. freq. for UFM is 21.34±9.00 kHz. This means that most UFM calls have the mean fundamental frequency lower 25 kHz, what is the criterion for separation between the low-frequency and high-frequency calls. Please substantiate, why you assign all calls of this type to the high-frequency calls. It seems that, in this study you conduct the border between the low-frequency and high-frequency calls at 20 kHz.

L 3 between the four pairs.

Please replace “four pairs” with four tests.

L 16-17 In addition, there was a negative correlation between the numbers of high- and low-frequency calls

Please add for the duration of a test after calls

Fig 4. Numbers of two types of calls by pairs during resident-intruder tests.

Please consider re-writing to “Numbers of two call types by resident-intruder pairs.

Relationship between vocalizations and aggressive interactions during the same-sex resident-intruder tests

L. 2-3. Relationship between vocalizations and aggressive interactions during the same-sex resident-intruder tests

Please re-write this heading to make it clear for the reader. What relationship you look, during (along) the test or between tests?

L. 4-5. We then examined the relationship between vocalizations and behaviors, with an emphasis on aggressive behaviors.

Delete, these are not the Results

L. 5 First, we created the ethograms for each pair

You use “pair” for designations of particular tests (one of 26), and also for designations of one of the four test categories (MF-VF etc.). This makes understanding the text impossible. Please correct terminology throughout the MS.

L. 5-7 First, we created the ethograms for each pair (examples shown in Fig. 5A) and categorized the behaviors into two categories, non- agonistic or agonistic (indicated by blue or red in Fig. 5A). Then, we quantified the type of behavior at the timing of high- or low-frequency calls emitted

This is repetition (first sentence) and amendments to Methods. If the authors provide in the Results something that was not indicated in the Methods, this should be transferred to Methods. How the authors confronted type of behavior at the timing of high- or low-frequency calls emitted? Behaviour was recorded by video, whereas the calls by UltraSoundGate in the form of ultrasonic files. How it was synchronized? Please described this in detain in the Methods.

L. 8 Replace Fig. 5B with Fig. 5A.

Discussion

P.21 L. 24-26 Furthermore, we defined males who experienced both mating and parenting as sexually experienced. If parenting is not a part of the criteria in other studies, this might cause differences between the previous and present findings.

Was it really part of your results? At least, it is lacking in the aim of MS.

L 9-12 bank voles [33] and Syrian hamsters .... Siberian hamsters [36].

Add Latin names

P. 22 L. 20-21 These nine calls are categorized into two types; high-frequency vocalization and low-frequency vocalization with harmonics.

This belongs to Methods, delete this.

L 3 California mice [41].

Add Latin name

L 19-21 These correlations strongly suggest that low-frequency calls would represent aggression or serve as a threat and warning consistent with the idea implied in previous reports [17, 18].

This conclusion is not supported with results of this study. Indeed, you do not know who emits the low-frequency calls, aggressor or defender. Delete this.

L 26-01 Low-frequency calls emitted in MF-VF gerbil interactions would be a good characteristic behavior to study multiple aspects of the expression of aggression in rodents. In addition, there is a possibility that low-frequency vocalizations could be linked to not only aggression but also important characteristics of a vocalizer, such as sex, reproductive state, or social experiences.

Again, your data do not show which animal is vocalizer, the winner or defender. During encounters of rodents, as a rule, the defender produces the human-audible (below 20 kHz) calls. The defender attacks silently.

In rats, human-audible (below 20 kHz) calls are emitted by the defending individuals during agonistic interactions (squeal, Watts, 1980), during tail-clamp (Chen et al. 2017) and in response to electrical nociceptive stimuli (Jourdan et al., 1995).

Watts, 1980. Vocalizations of nine species of rat (Rattus; Muridae). J. Zool., 191:531-555.

Chen et al., 2017. Call divergence in three sympatric Rattus species. J. Acoust. Soc. Am. 142:29-34.

Jourdan et al., 1995. Audible and ultrasonic vocalization elicited by single electrical nociceptive stimuli to the tail in the rat. Pain, 63:237-249.

L 6-8 MFs (resident) most likely emitted low-frequency calls during the resident-intruder test since low-frequency calls were rarely observed in VF-VF interactions

Please indicate in Methods in detail, how did you determine, who emitted the low-frequency calls during the resident-intruder test. This is important thing, as your conclusions that low-frequency calls belong to aggressor rather than defender contradict to data by Ter-Mikaelian et al. 2012 (cited in MS) on Mongolian gerbils, as well as with data on other species of gerbils (see below).

Volodin et al., 1994. Situational changes in vocalization of Great gerbils (Rhombomys opimus Licht.) during defensive behavior. Doklady Biological Sciences, 334:65-68.

Volodin I.A., Goltsman M.E., 2000. Acoustic activity displayed in the agonistic behavior of Great and Light gerbils. Doklady Biological Sciences, 371:176-178.

L 10-12 However, given that low-frequency calls were recorded even during the period before the resident started attacking the intruder, it would be reasonable to consider that the majority of low-frequency calls were emitted by MF residents.

This is unsupported claim. The animal introduced on the territory of resident afraid of it and start calling.

Reviewer #2: Comments to the Author

Behaviors and vocalizations associated with aggression are essential for animals

to survive, reproduce, and maintain their community. This study reported that high- and low-frequency vocalizations relate to non-agonistic and agonistic interactions during encounters in Mongolian gerbils, respectively, which related to the sexual experience. These findings provide new insights into the modulatory effects of sex and sexual experience on vocalizations during agonistic encounters.

I have some major comments:

1) Why do intruders only use virgin gerbils ？

2) Whether or not there are difference between the structure of vocalizations before the attack and the after the attack? It might be possible that fighting has been expressed in voice before the beginning of physical conflict, which is a characteristic of vocalizations in the aggressive strategy.

3) It should be added to some explanations on the ecological function of the high- and low-frequency vocalizations related to non-agonistic and agonistic interactions in the discussion.

The minor comments:

1) Is it more appropriate to change the title to“ Vocalization during agonistic encounter in Mongolian gerbils：impact of sexual experience.”

2) Result Lines 3-7 should belong to the method behavior observation part，that is part of“same-sex resident-intruder test” , and it is recommended to adjust.

3) Page 23 line 13 “These results indicate that vocalizations….” should revised” These results indicated that vocalizations….”

Reviewer #3: This is a fantastic publication exploring not only vocalization production in Mongolian gerbils but also how sex and sexual experience effects aggression and the expression of these calls. I have a few minor points and questions to be addressed, some areas that require clarification or a better explanation, my comments are below.

Abstract

Line 3 – maintain their community, does aggression do this? Or would it be more appropriate to say maintain their social hierarchy?

Line 12 – perhaps spilt these sentences

Introduction

Line 5 – define clinching, and make sure the references cover this behaviour (1,2 don’t cover clinching, I couldn’t find clinching in 3, but of it is defined in reference 3 then ignore this comment)

Methods

Page 1 – line 29 what software was used to score the videos?

Page 2 – line 1-4 a table describing these behaviours would be helpful. Additionally, are these behaviours arbitrarily selected or have they been previously used/described?

Page 3 – If the calls were selected using a MATLAB script was this program 100% accurate at distinguishing calls? Also was this checked by an experimenter? Further, were the calls classified by an experimenter or was this also performed by the MATLAB script?

Page 4 – chi square symbol isn’t showing up

Page 4 – please explain why a non-parametric test was used instead of an ANOVA

Results

Page 1 - line 3 – in the gerbils

Page 2 – line 18 – nine calls of vocalizations should be nine types of vocalizations

Page 3 – line 1 – why were the calls grouped in those parameters?

Discussion

Page 1 – line 20 – territorial sentence could be re-worded so it flows better

Page 2 – line 24 – rats actually make 50-khz USV when they are in aggressive situations as well as play (but 22kHz calls are fairly unique to aggression, just like what you saw!) see Burke et al., 2017 (Avoiding escalation from play to aggression in adult male rats: The role of ultrasonic calls)

Page 3 – I think that these are great points, again the rat literature really mimics your findings so potentially a comparison to this literature would really make your point a bit stronger (recent publications from Pellis/Burke; Wohr/Schwarting/Kisko) This is just a suggestion, and is not at all necessary for the publication.

6. PLOS authors have the option to publish the peer review history of their article (what does this mean?). If published, this will include your full peer review and any attached files.

Reviewer #1: No

Reviewer #2: No

Reviewer #3: No

---

## [Author Response · Author response to Decision Letter 0]

18 Mar 2022

Editor

As detailed below, we have addressed all of the reviewer’s reservations.

Here are responses to comments which are not written in the reviewer’s comments.

>It seems to me you used the subjects repeatedly on the tests. If not, you should clearly say it. 

In the original text, this is clearly stated that “Each subject (resident and intruder) underwent the resident-intruder test for a single time.”. 

>Your analysis did not show the interrelationships between the factors, such as the age of the subjects, their sexual experience, the number of animals kept in one plexiglass cage, the latency and duration of aggressive behaviours, etc.

The age of animals might have some effect on the behaviors, however, we did not conduct enough experiments to examine the effect. We raised the gerbils under the guideline which is internationally well accepted. We think that usually this factor “the number of animals kept in one plexiglass cage” is not considered in this kind of experiment. For the rest, we compared by using no-parametric tests. We used no-parametric tests, since the acquired data did not show normal distributions.

>The uncategorized correlation in Fig. 2 is misleading. You present a negative correlation. However, it seems the case for MF-VF, not that much for EM-VM, and it is unlikely for the rest of the categories. (Similarly, in Fig. 6 B, C, and D.) 

We tested the correlation for each subgroup and there was no statistical significance. We removed the sentence emphasizing the correlation from the abstract and discussion.

Reviewer #1

Thank you for your kind suggestions to improve this manuscript.

As detailed below, we have addressed all of the reviewer’s reservations.

The changes are highlighted in the marked text and “LXX” corresponds to lines in the marked text.

Abstract

L. 9-10 We also confirmed two types of vocalisations: high-frequency (>25 kHz) and low-frequency (<25 kHz)

This is strange subdivision. Commonly, classifying calls in the Mongolian gerbil is to ultrasonic calls over 20 kHz and human-audible calls below 20 kHz.

Author’s response

We agree that the separation at 25 kHz seems arbitral and not common. However, we think that the separation human-audible or not is also an arbitral separation. Then, we calculated the fitting functions of two normal distributions for high- and low- vocalization distributions. The intersection of the two functions is used as the objective frequency border between the high-frequency and the low-frequency vocalizations. The number was “24.6 kHz”. This is now explained in the method section (L176-182).

L. 10 with multiple harmonics

This has not sense. Harmonics are integer multiples of fundamental frequency (the lowest frequency band). If the fundamental frequency is low, they are spaced more densely, if the fundamental frequency is high, they are spaces more rarely. So, you just do not see the harmonics of high-frequency calls.

Author’s response

We think that this explanation will help to grasp the character of the vocalization, for the readers not familiar with sound physics.

L. 10-14

The call rate of high-frequency vocalizations was negatively correlated with the duration of aggressive behavior, while the call rate of low-frequency vocalizations was negatively correlated with the latency of aggressive behavior and positively correlated with the duration of aggressive behavior.

Who vocalized during the aggressive encounters: the victim, the victor, both fighters?

Author’s response

We did not specify the caller, because of technical limitations. Probably both emitted the calls.

We changed the sentence to “At the timing of high-frequency vocalizations observed during the tests, the vast majority (96.2%) of the behavioral interactions were non-agonistic. While at the timing of low-frequency vocalizations observed during the tests, around half (45%) of the behavioral interactions were agonistic.” (L31-35).

Introduction

L. 14-15 As it is well known, laboratory rodents such as mice or rats rarely exhibit this extent of aggression.

This statement needs in references, please provide them.

Author’s response

We added references (L56-57).

L. 1-6. Although several studies cited in the previous paragraph have reported aggression-specific vocalizations, it remains unclear how sex and sexual experience affect the call rate of various vocalizations in Mongolian gerbils. Thus, we analyzed vocalizations between four different pairs (virgin male–virgin male; experienced male– virgin male; virgin female–virgin female; multiparous female–virgin female) of Mongolian gerbils through the same-sex resident-intruder test.

The aim of the study is poorly formulated and does not reflect the content of the paper. Did you analysed vocalizations, call rates and behaviour latencies? For what? Please re-write the aim accordingly to you real results.

Author’s response

We modified the paragraph (L71-78).

Methods

Same-sex resident-intruder test

L 16 We conducted resident-intruder tests

Добавьте 26 перед resident-intruder tests

Author’s response

We added the number (L95).

L 29 - 04 The observed behaviors were classified as follows: alert posture, ano-genital sniff, approach, attack, clinch, chase, dig, explore, fight, flee, jump, move away, nasal sniff, push, self-groom, stop moving, and watch. ‘attack’, ‘chase’, ‘clinch’, ‘flee’, and ‘push’ were scored as aggressive/agonistic behaviors. The rest was scored as non-agonistic interaction.

It is necessary to provide the references to papers (they are numerous) and/or short descriptions of the forms of aggressive and non-aggressive behaviour of rodents, to make clear for the reader, on what you based your separation to the aggressive/agonistic and non-agonistic behaviors. Why fight is not assigned to the aggressive/agonistic behaviors?

Author’s response

Thank you for pointing it out. We added a reference with which we made the classification and made a table to explain each behavior (L129). Also, we fixed the term lists for the aggressive behaviors (L111-116).

Recording and analysis of vocalizations

L. 24 Sonograms

Thereafter you use more correct term “spectrogram”. Use it here too.

Author’s response

We replaced the word ‘sonograms’ with ‘spectrogram’ (L142).

L 1 Delete total duration after F0

Author’s response

‘total duration’ was deleted (L147).

L 16-17 Thus, we identified nine calls in this paper; U-shape, UFM-s, UFM, AFM, DFM-l, DFM, QCF, NB, and QCF-NB

Please provide here as an illustration Fig.3A as Fig.

Please make it clear, whether these nine call types could be only low-frequency or only high-frequency or both low- and high-frequency?

The authors should indicate how they classified the calls to the high- and low-frequency (for example, if F0max=30 kHz and F0min=20 kHz). Was it done based on visual inspection of spectrograms or measuring fundamental frequency?

Author’s response

We provided Fig.3A (now Fig.1) here (L184).

We categorized calls using their maximum fundamental frequency. Calls F0max ≥24.6 kHz were assigned to high-frequency calls and calls F0max <24.6 kHz were assigned to low-frequency calls (the calculation to determine this border was described above and now in the method section). No call was assigned into both high & low calls. (L176-182)

Statistical analysis

L 8-9 number of high-frequency vocalizations per minute; latency of aggressive behavior and the number of low-frequency vocalizations per minute;

Please explain in the previous section, how did you measure the number of high-frequency vocalizations per minute and number of low-frequency vocalizations per minute. From the entire test duration or only from the duration after the first aggression?

Author’s response

Vocalization per minute was calculated from the entire test session.

We added explanations to make this clear (L192-195).

Results

Same-sex resident-intruder test

L 3-4 We conducted resident-intruder tests to assess the extent of territorial aggression in gerbils. All intruders were virgin and of the same sex as the resident.

Delete, these are not the results

Author’s response

We deleted these (L206-210).

L 4-7 Many of the residents expressed aggressive behaviors toward the intruder for several minutes and frequently emitted calls. The vast majority of aggressive behaviors were initiated by residents, thus basically we focused on the behaviors of residents in this study.

Please decode what are Many of the residents and The vast majority. In the Results, you should provide the digits, not judgements.

Author’s response

We provided the actual numbers, and this paragraph was moved to the method section (L104-107).

L 4-5 Many of the residents expressed aggressive behaviors toward the intruder for several minutes and frequently emitted calls

How the authors determined who is calling, the resident or the intruder? In gerbils, both can call. The low-frequency calls produce the fleeing animals during aggressive contacts (Ter-Mikaelian et al. 2012, cited in MS). This is a key question, which should be addressed in the Methods.

Author’s response

We did not specify the caller. We deleted this sentence about calls (L104-107). Sorry for the confusion.

Analysis of vocalizations of gerbils during the same-sex resident-intruder tests

L. 14-15. Analysis of vocalizations of gerbils during the same-sex resident-intruder tests

Delete “Analysis of”. Here, you describe the results of analysis rather than analysis itself.

Author’s response

We deleted this (L248).

L 16 Gerbils frequently emitted calls during the resident-intruder test

Delete this sentence

Author’s response

We deleted this (L250).

L. 18. We detected nine calls of vocalizations

This sounds senseless. Do you mean “we detected nine call types”?

Author’s response

We rewrite this as “We detected nine call types of vocalizations”.

L 20-23 The spectral shapes of calls (Fig. 3A; except the U-shape call, which was newly defined in this study) in all pairs during the resident-intruder test were similar to those reported in previous studies [17, 18]. Alert calls described in the same study [17] were not observed in our resident-intruder tests.

This belongs to Discussion. Data from previous studies should not be discussed in the Results section.

Author’s response

We moved this to the discussion with modification (L257, L418-422).

L. 20 spectral shapes

This sounds unclear and imprecise. Replace with “contour shapes”.

Author’s response

We replaced this (L254).

L. 17-18 It has been reported that the frequency of vocalizations bears semantic significance with a dividing line at 25 kHz [17, 18]. Vocalizations with a maximum fundamental frequency above 25 kHz were observed during non-agonistic interactions, whereas those below 25 kHz were observed during agonistic interactions. The separation at 25 kHz is consistent with our observations (Fig. 3B).

This part of text represents a mix of Methods, own Results and Discussion. It should be separated accordingly to these sections. Please substantiate the separation to the low-frequency and high-frequency calls in Introduction. Please provide the criteria for separation to these categories in Methods. Otherwise, Statistical analysis remains perfectly unclear.

Author’s response

We added the criteria to separate high- and low-frequency calls in the method section (L176-182).

Also, part of this section was moved to methods and discussions.

L. 3-5 high-frequency with short duration (U-shape, UFM-s, and UFM; 32.7 ± 2.0 kHz, 25.9 ± 14.7 ms), and low-frequency with long duration (AFM, DFM-l, DFM, QCF, NB, and QCF-NB; 12.1 ± 2.9 kHz, 160.4 ± 73.3 ms)

Please indicate that the fundamental frequency (maximal?) and the duration are provided for all high-frequency and low-frequency криков. Please indicate the numbers of the high-frequency and low-frequency calls.

Author’s response

F0 for each call is shown in table 2. The numbers of the high-frequency (3329) and low-frequency calls (1033) are now indicated (L268-270).

Fig. 3 legend

Replace “submitted to” with “in”

Author’s response

We replaced “submitted to” with “during” (L272).

Fig. 3 legend MF-VF pairs seemed to have a different composition of syllables from the other pairs.

Transfer this from the figure legend to main text.

Author’s response

We deleted this sentence (L277).

Table 1. Spectro-temporal features of nine calls.

Replace “nine calls” “nine call types” here and in the sentence under the Table 1. Please add to the legend designations/ decoding of all acoustic variables presented in the column headings.

Author’s response

We followed your suggestions (L279).

Table 1. Spectro-temporal features of nine calls.

Please indicate n (call number) for each call type

Author’s response

Now those are indicated in table 2 (L280-285).

Table 1. UFM

Mean Min. freq. for UFM is 21.34±9.00 kHz. This means that most UFM calls have the mean fundamental frequency lower 25 kHz, what is the criterion for separation between the low-frequency and high-frequency calls. Please substantiate, why you assign all calls of this type to the high-frequency calls. It seems that, in this study you conduct the border between the low-frequency and high-frequency calls at 20 kHz.

Author’s response

We categorized calls using their maximum fundamental frequency. Calls F0max ≥24.6 kHz were assigned to high-frequency calls and calls F0max <24.6 kHz were assigned to low-frequency calls (the calculation to determine this new border was described above and now in the method section; L176-182).

L 3 between the four pairs.

Please replace “four pairs” with four tests.

Author’s response

We replaced “four pairs” with “four tests” (L288).

L 16-17 In addition, there was a negative correlation between the numbers of high- and low-frequency calls

Please add for the duration of a test after calls

Fig 4. Numbers of two types of calls by pairs during resident-intruder tests.

Please consider re-writing to “Numbers of two call types by resident-intruder pairs.

Author’s response

We rewrote as suggested (L305).

Relationship between vocalizations and aggressive interactions during the same-sex resident-intruder tests

L. 2-3. Relationship between vocalizations and aggressive interactions during the same-sex resident-intruder tests

Please re-write this heading to make it clear for the reader. What relationship you look, during (along) the test or between tests?

Author’s response

We rewrote the heading as “Relationship between each vocalization and aggressive interactions during the same-sex resident-intruder tests” (L314).

L. 4-5. We then examined the relationship between vocalizations and behaviors, with an emphasis on aggressive behaviors.

Delete, these are not the Results

Author’s response

The sentence was deleted (L316-317).

L. 5 First, we created the ethograms for each pair

You use “pair” for designations of particular tests (one of 26), and also for designations of one of the four test categories (MF-VF etc.). This makes understanding the text impossible. Please correct terminology throughout the MS.

Author’s response

We changed “pairs” to “pair-tests” in some text which might cause confusion. For this sentence, “each pair” was replaced with “26 tests” (L317).

L. 5-7 First, we created the ethograms for each pair (examples shown in Fig. 5A) and categorized the behaviors into two categories, non- agonistic or agonistic (indicated by blue or red in Fig. 5A). Then, we quantified the type of behavior at the timing of high- or low-frequency calls emitted

This is repetition (first sentence) and amendments to Methods. If the authors provide in the Results something that was not indicated in the Methods, this should be transferred to Methods. How the authors confronted type of behavior at the timing of high- or low-frequency calls emitted? Behaviour was recorded by video, whereas the calls by UltraSoundGate in the form of ultrasonic files. How it was synchronized? Please described this in detain in the Methods.

Author’s response

We think this explanation will help readers to understand the flow, even it is containing a kind of method. The timing was synchronized with the sound in the video and the audio file. We added this explanation in the method section (L181-182).

L. 8 Replace Fig. 5B with Fig. 5A.

Author’s response

We think the original is correct.

Discussion

P.21 L. 24-26 Furthermore, we defined males who experienced both mating and parenting as sexually experienced. If parenting is not a part of the criteria in other studies, this might cause differences between the previous and present findings.

Was it really part of your results? At least, it is lacking in the aim of MS.

Author’s response

This is the discussion to speculate the difference of results between the reports including ours.

L 9-12 bank voles [33] and Syrian hamsters .... Siberian hamsters [36].

Add Latin names

Author’s response

We added these Latin names “Myodes glareolus”, “Mesocricetus auratus”, and “Phodopus sungorus” (L407,410).

P. 22 L. 20-21 These nine calls are categorized into two types; high-frequency vocalization and low-frequency vocalization with harmonics.

This belongs to Methods, delete this.

Author’s response

We think that this sentence will help to understand the flow.

L 3 California mice [41].

Add Latin name

Author’s response

We added “Peromyscus acalifornicus” (L439).

L 19-21 These correlations strongly suggest that low-frequency calls would represent aggression or serve as a threat and warning consistent with the idea implied in previous reports [17, 18].

This conclusion is not supported with results of this study. Indeed, you do not know who emits the low-frequency calls, aggressor or defender. Delete this.

Author’s response

We deleted this.

L 26-01 Low-frequency calls emitted in MF-VF gerbil interactions would be a good characteristic behavior to study multiple aspects of the expression of aggression in rodents. In addition, there is a possibility that low-frequency vocalizations could be linked to not only aggression but also important characteristics of a vocalizer, such as sex, reproductive state, or social experiences.

Again, your data do not show which animal is vocalizer, the winner or defender. During encounters of rodents, as a rule, the defender produces the human-audible (below 20 kHz) calls. The defender attacks silently.

In rats, human-audible (below 20 kHz) calls are emitted by the defending individuals during agonistic interactions (squeal, Watts, 1980), during tail-clamp (Chen et al. 2017) and in response to electrical nociceptive stimuli (Jourdan et al., 1995).

Watts, 1980. Vocalizations of nine species of rat (Rattus; Muridae). J. Zool., 191:531-555.

Chen et al., 2017. Call divergence in three sympatric Rattus species. J. Acoust. Soc. Am. 142:29-34.

Jourdan et al., 1995. Audible and ultrasonic vocalization elicited by single electrical nociceptive stimuli to the tail in the rat. Pain, 63:237-249.

L 6-8 MFs (resident) most likely emitted low-frequency calls during the resident-intruder test since low-frequency calls were rarely observed in VF-VF interactions

Please indicate in Methods in detail, how did you determine, who emitted the low-frequency calls during the resident-intruder test. This is important thing, as your conclusions that low-frequency calls belong to aggressor rather than defender contradict to data by Ter-Mikaelian et al. 2012 (cited in MS) on Mongolian gerbils, as well as with data on other species of gerbils (see below).

Volodin et al., 1994. Situational changes in vocalization of Great gerbils (Rhombomys opimus Licht.) during defensive behavior. Doklady Biological Sciences, 334:65-68.

Volodin I.A., Goltsman M.E., 2000. Acoustic activity displayed in the agonistic behavior of Great and Light gerbils. Doklady Biological Sciences, 371:176-178.

L 10-12 However, given that low-frequency calls were recorded even during the period before the resident started attacking the intruder, it would be reasonable to consider that the majority of low-frequency calls were emitted by MF residents.

This is unsupported claim. The animal introduced on the territory of resident afraid of it and start calling.

Author’s response

First of all, we have not determined which gerbil emitted the calls. 

These discussions are rational speculations, not definite conclusions. Here we have raised possibilities to be tested with more elaborate experiments in the future. 

The reviewer claimed that “During encounters of rodents, as a rule, the defender produces the human-audible (below 20 kHz) calls”. However, this is not a general rule. Occasionally, offending rats emit 22 kHz calls during the encounter as reported in Burke et al. 2017. Also, Morton has claimed that many animals including rodents use low-frequency vocalizations when they are hostile. We agree that aversive stimuli trigger 22 kHz calls in rats and defeated rats tend to emit 22 kHz calls. Meanwhile, many reports also suggested that rats emit 22 kHz calls in alert situations, the proximity of threats or presentation of predator signals (“Handbook of Ultrasonic Vocalization” edited by Brudzynski S.M.; published from Academic Press). In our experiment, resident gerbils could be the aggressive ones and also the ones who perceive threats from the outside. The territorial aggression could be also a response to threat, not only a simple expression of aggression. 

Furthermore, Ter-Mikaelian and colleagues did not conclude as the reviewer wrote. I quote the corresponding paragraph here.

“3.2.2. Vocalization characteristics

Aggression calls were emitted during the watch posture, nasal sniff, sidling, or when the submissive animal began to flee. Since the vocalizations were heard when there was no body contact, they probably did not signal pain. It was not possible to determine with certainty which animal emitted the vocalizations during aggressive encounters; the calls may signal submission by the weaker animal or aggression by the dominant animal. However, on certain occasions, it was noted that the fleeing animal emitted vocalizations.”

The authors suggest that this kind of low-frequency call relates to aggression and submission, and clearly state that they were not able to determine the caller.

The experiments conducted by Volodin and colleagues also did not determine the caller, and used the other species of gerbils.

Our results indicated that a considerably large number of low-frequency calls were emitted in the tests MF involved compared with the other 3 tests. It is reasonable to estimate MFs will have different characters than others. Also, Low-frequency calls were confirmed before the first agonistic event in the MF-VF tests. If the intruders emit low-frequency calls because they are afraid of residents, low-frequency calls should be recorded before the first agonistic event in all 4 tests, but this was not the case. Taken together, our estimation that the MFs might emit low-frequency calls must be rational enough.

 

Reviewer #2

Thank you for your kind suggestions to improve this manuscript.

As detailed below, we have addressed all of the reviewer’s reservations.

The changes are highlighted in the marked text and “LXX” corresponds to lines in the marked text.

The major comments: 

Reviewer’s comment 1: Why do intruders only use virgin gerbils?

Author’s response: Since we intended to mainly observe the territorial aggression of residents, virgin gerbils were used as intruders. If the experienced subjects were used as intruders, some of them would exhibit aggressive behaviors actively towards residents. In this experiment, we tried avoiding this.

Reviewer’s comment 2: Whether or not there are difference between the structure of vocalizations before the attack and the after the attack? It might be possible that fighting has been expressed in voice before the beginning of physical conflict, which is a characteristic of vocalizations in the aggressive strategy.

Author’s response: Thank you for your suggestion. We tested this and the result is shown in the L335-347. In brief, there were some modulations both in high- and low-frequency calls.

Reviewer’s comment 3: It should be added to some explanations on the ecological function of the high- and low-frequency vocalizations related to non-agonistic and agonistic interactions in the discussion.

Author’s response: We add discussion sentences about the ecological function (L423-427).

The minor comments:

Reviewer’s comment 1: Is it more appropriate to change the title to“ Vocalization during agonistic encounter in Mongolian gerbils: impact of sexual experience.”

Author’s response: Thank you for your suggestion. We changed the title.

Reviewer’s comment 2: Result Lines 3-7 should belong to the method behavior observation part, that is part of“same-sex resident-intruder test” , and it is recommended to adjust.

Author’s response: We moved the sentences to the method section and adjusted (L104-107).

Reviewer’s comment 3: Page 23 line 13 “These results indicate that vocalizations….” should revised” These results indicated that vocalizations….”

Author’s response: We changed the tense as pointed (L486).

 

Reviewer #3

Thank you for your kind suggestions to improve this manuscript.

As detailed below, we have addressed all of the reviewer’s reservations.

The changes are highlighted in the marked text and “LXX” corresponds to lines in the marked text.

Reviewer’s comment 1

Abstract

Line 3 – maintain their community, does aggression do this? Or would it be more appropriate to say maintain their social hierarchy?

Line 12 – perhaps spilt these sentences

Author’s response: We modified abstract L3. Please take a look at L24.

The phrase “maintain their community” is replaced with “organize their social hierarchy”

For line 12, we followed your suggestion and modified the sentence (L32-35).

Reviewer’s comment 2

Introduction

Line 5 – define clinching, and make sure the references cover this behaviour (1,2 don’t cover clinching, I couldn’t find clinching in 3, but of it is defined in reference 3 then ignore this comment)

Author’s response: “clinching” was not an appropriate term. We replaced “clinching” with “boxing” in the manuscript. L46, L114

Reviewer’s comment 3

Methods

Page 1 – line 29 what software was used to score the videos?

Author’s response: We manually scored the behaviors by watching the videos. The method was modified to make this clear (L113).

Page 2 – line 1-4 a table describing these behaviours would be helpful. Additionally, are these behaviours arbitrarily selected or have they been previously used/described?

Author’s response: These behaviors are selected by following the previous report (Ter-Mikaelian et al. 2012) with a small modification. The reference was added in this sentence (L111-112). A table was added (L129).

Page 3 – If the calls were selected using a MATLAB script was this program 100% accurate at distinguishing calls? Also was this checked by an experimenter? Further, were the calls classified by an experimenter or was this also performed by the MATLAB script?

Author’s response: Each vocalization was detected and cut out from the audio files manually by experimenters. The classification was also manually conducted by experimenters following the characters described in the ref 17. These explanations were added in the method section (L138-139).

Page 4 – chi square symbol isn’t showing up

Author’s response: We fixed this (L187).

Page 4 – please explain why a non-parametric test was used instead of an ANOVA

Author’s response: Since the acquired data did not show the normal distribution, we used the non-parametric tests.

Reviewer’s comment 4

Results

Page 1 - line 3 – in the gerbils

Author’s response: We changed as pointed out. The sentence was moved in the method section (L211).

Page 2 – line 18 – nine calls of vocalizations should be nine types of vocalizations

Author’s response: We replaced the word “calls” with “call types” (L252).

Page 3 – line 1 – why were the calls grouped in those parameters?

Author’s response: Now, an objective parameter was calculated. We calculated the fitting functions of two normal distributions for high- and low- vocalization distributions. The intersection of the two functions is used as the objective frequency border between the high-frequency and the low-frequency vocalizations. The number was “24.6 kHz”. This is explained in the method section (L176-182).

Reviewer’s comment 5

Discussion

Page 1 – line 20 – territorial sentence could be re-worded so it flows better

Author’s response: We rewrite the sentence as, “Mongolian gerbils exhibit territorial aggression in a considerably large territory [11, 12].” (L387-388).

Page 2 – line 24 – rats actually make 50-khz USV when they are in aggressive situations as well as play (but 22kHz calls are fairly unique to aggression, just like what you saw!) see Burke et al., 2017 (Avoiding escalation from play to aggression in adult male rats: The role of ultrasonic calls)

Author’s response: We added that rats emit USV in an aggressive situation (L430).

Page 3 – I think that these are great points, again the rat literature really mimics your findings so potentially a comparison to this literature would really make your point a bit stronger (recent publications from Pellis/Burke; Wohr/Schwarting/Kisko) This is just a suggestion, and is not at all necessary for the publication.

Author’s response: Thank you for the constructive suggestion! The literature listed are quite encouraging us. We add a comparison in the discussion (L453-458).

---

## [Decision Letter · Decision Letter 1]

12 May 2022

PONE-D-21-38093R1Vocalization during agonistic encounter in Mongolian gerbils: impact of sexual experience.PLOS ONE

Dear Dr. Yamamoto,

Thank you for submitting your manuscript to PLOS ONE. After careful consideration, we feel that it has merit but does not fully meet PLOS ONE’s publication criteria as it currently stands. Therefore, we invite you to submit a revised version of the manuscript that addresses the points raised during the review process.

I’m sorry you haven’t received a decision yet. We have a somewhat delayed promised third review on your revision. It takes time to write a qualified review. If I have a forthcoming review, I prefer a delayed decision to the disappointment of a volunteer opponent who has already invested her/his time in reading the manuscript. However, I agree that the matter is already considerably delayed. I am sending you my decision without a third review.

As you see, two reviewers are quite happy with your revision. However, they suggested some minor changes, Reviewer 1 in particular. If you changed the details as instructed, I would accept it. Please submit your revised manuscript by Jun 26 2022 11:59PM. If you will need more time than this to complete your revisions, please reply to this message or contact the journal office at plosone@plos.org. Please include the following items when submitting your revised manuscript:A rebuttal letter that responds to each point raised by the academic editor and reviewer(s). You should upload this letter as a separate file labeled 'Response to Reviewers'.A marked-up copy of your manuscript that highlights changes made to the original version. You should upload this as a separate file labeled 'Revised Manuscript with Track Changes'.An unmarked version of your revised paper without tracked changes. You should upload this as a separate file labeled 'Manuscript'.If applicable, we recommend that you deposit your laboratory protocols in protocols.io to enhance the reproducibility of your results. Protocols.io assigns your protocol its own identifier (DOI) so that it can be cited independently in the future. For instructions see: https://journals.plos.org/plosone/s/submission-guidelines#loc-laboratory-protocols. Additionally, PLOS ONE offers an option for publishing peer-reviewed Lab Protocol articles, which describe protocols hosted on protocols.io. Read more information on sharing protocols at https://plos.org/protocols?utm_medium=editorial-email&utm_source=authorletters&utm_campaign=protocols.

We look forward to receiving your revised manuscript.

Kind regards,

Ludek Bartos

Academic Editor

PLOS ONE

Journal Requirements:

Reviewers' comments:

Reviewer's Responses to Questions

**Comments to the Author**

1. If the authors have adequately addressed your comments raised in a previous round of review and you feel that this manuscript is now acceptable for publication, you may indicate that here to bypass the “Comments to the Author” section, enter your conflict of interest statement in the “Confidential to Editor” section, and submit your "Accept" recommendation.

Reviewer #1: (No Response)

Reviewer #2: All comments have been addressed

2. Is the manuscript technically sound, and do the data support the conclusions?

Reviewer #1: Yes

Reviewer #2: Yes

3. Has the statistical analysis been performed appropriately and rigorously? 

Reviewer #1: Yes

Reviewer #2: Yes

4. Have the authors made all data underlying the findings in their manuscript fully available?

Reviewer #1: (No Response)

Reviewer #2: Yes

5. Is the manuscript presented in an intelligible fashion and written in standard English?

Reviewer #1: (No Response)

Reviewer #2: Yes

6. Review Comments to the Author

Reviewer #1: The authors addressed most of my comment and the revised MS was substantially improved. However, there are some minor points which should be resolved before publishing, as they confuse the reader.

L 30. We also confirmed two types of vocalizations during the encounters

Replace "two types" with "two groups of vocalizations" (as in L 176). Otherwise, types of vocalizations confuse with nine call types (see also L 256 and L 405).

L 31. high-frequency (>24.6 kHz) and low-frequency (<24.6 kHz) with multiple harmonics

Delete "with multiple harmonics" after "low-frequency". The high-frequency vocalizations also have multiple harmonics. Create the spectrum up to 300 kHz, increase the intensity, and you will see them clearly.

L 106 Aggressive behaviors were initiated by residents in the vast majority (20/24) of tests

Why 24, not 26? You have in total 26 tests.

L 163. we identified nine calls

Replace "nine calls" with "nine call types"

L 224 gerbil groups submitted to resident-intruder tests.

Replace "submitted to" with "during".

L 254-256 the nine call types could be categorized into two groups. From the fitting functions of two normal distributions, the separation at 24.6 kHz was determined. Thus, for further analysis, we categorized the quantified nine calls of vocalizations into two types

Nine call types (L 254) cannot be categorized to the two call types. Replace with "we categorized the nine types of vocalizations into two groups"

L 262-263 High-frequency syllables are represented by bluish symbols and low-frequency syllables

Replace "syllables" with "groups" (as earlier in L 176). Avoid synonyms.

L 402-406 We found that gerbils emitted nine types of vocalizations during the resident-intruder test. The spectral shapes of these calls were basically consistent with those reported in previous studies [17, 18]. Alert calls described in the same study [17] were not observed in our resident-intruder tests. The nine calls are categorized into two types; high-frequency vocalization and low-frequency vocalization with harmonics.

Nine types of vocalizations (L 402) cannot be categorized to the two call types. Using consistently own terms throughout the text is important for understanding the content by the readers. Replace with "The nine call types are categorized into two groups"

L 406 low-frequency vocalization with harmonics.

Delete "with harmonics". Presence of visible harmonics is not exclusively attributive to low-frequency vocalizations, but depends on the size of the used for analysis spectral window.

L 444 resident intruder

Replace with resident-intruder

L 443-456 This paragraph do not provides useful information and is very far from the results of this study. Delete it.

L 464-467 However, given that low-frequency calls were recorded even during the period before the resident started attacking the intruder in MF-VF pairs (this was not observed in the other groups), it would be reasonable to estimate that the majority of low-frequency calls were emitted by MF residents.

This not an argument. Avoid speculations. In rodents, a lot of information comes from olfactory channel. It is easy to propose that VF-intruders placed on territory of MF-residents, perceive the smell of adult females and start calling of fear before the first aggressive interaction. A few more opposite arguments can be advanced.

L 473-476 However, it remains unknown why MF gerbils (or MF-VF pair) emitted more low-frequency calls compared with those in the other three pairs. We hypothesize that vocalizations are influenced by hormonal changes induced by parenting experiences in female Mongolian gerbils.

In the preceding paragraph, you write that it is impossible to establish, who is calling the low-frequency calls, MF or VF. However, here you attribute the calls to one of females of the pair. Please re-write to make the content consistent with the text above.

Reviewer #2: This revised version has revised and supplemented the necessary information, and if the minor comments were revised, it could be considered to accept.

I think the descriptions of several behaviors that were scored should be more specific and detailed. We cannot define a target behavior with itself, for example, dig was defined by digging, and what is dig?.

Jump Jumping vertically

Move away Moving away from the another

Dig Digging beddings on the floor

Explore Exploring the cage

Stop Stop moving

Approach Approaching within one body length of another

Maybe can refer to the reference: Hurtado-Parrado C, Gonzalez CH, Moreno LM, Gonzalez CA, Arias M et al., 2015. Catalogue of the behaviour of Meriones unguiculatus f. dom. (Mongolian gerbil) and wild conspecies, in captivity and under natural conditions, based on a systematic literature review. J Ethol 33:65–86

7. PLOS authors have the option to publish the peer review history of their article (what does this mean?). If published, this will include your full peer review and any attached files.

Reviewer #1: No

Reviewer #2: No

---

## [Author Response · Author response to Decision Letter 1]

16 May 2022

Reviewer #1

Thank you for your kind suggestions to improve this manuscript.

As detailed below, we have addressed all of the reviewer’s reservations.

The changes are highlighted in the marked text and “LXX” corresponds to lines in the marked text.

Reviewer’s comment:

L 30. We also confirmed two types of vocalizations during the encounters

Replace "two types" with "two groups of vocalizations" (as in L 176). Otherwise, types of vocalizations confuse with nine call types (see also L 256 and L 405).

Author’s response: We corrected as the reviewer suggested (L30). 

Reviewer’s comment:

L 31. high-frequency (>24.6 kHz) and low-frequency (<24.6 kHz) with multiple harmonics

Delete "with multiple harmonics" after "low-frequency". The high-frequency vocalizations also have multiple harmonics. Create the spectrum up to 300 kHz, increase the intensity, and you will see them clearly.

Author’s response: We deleted "with multiple harmonics" (L31).

Reviewer’s comment:

L 106 Aggressive behaviors were initiated by residents in the vast majority (20/24) of tests

Why 24, not 26? You have in total 26 tests.

Author’s response: We corrected the number. The correct number is 20/26 (L106). Thank you for pointing this out.

Reviewer’s comment:

L 163. we identified nine calls

Replace "nine calls" with "nine call types"

Author’s response: We replaced it (L163).

Reviewer’s comment:

L 224 gerbil groups submitted to resident-intruder tests.

Replace "submitted to" with "during".

Author’s response: We replaced it (L224).

Reviewer’s comment:

L 254-256 the nine call types could be categorized into two groups. From the fitting functions of two normal distributions, the separation at 24.6 kHz was determined. Thus, for further analysis, we categorized the quantified nine calls of vocalizations into two types

Nine call types (L 254) cannot be categorized to the two call types. Replace with "we categorized the nine types of vocalizations into two groups"

Author’s response: We replaced it (L256).

Reviewer’s comment:

L 262-263 High-frequency syllables are represented by bluish symbols and low-frequency syllables

Replace "syllables" with "groups" (as earlier in L 176). Avoid synonyms.

Author’s response: We replaced it (L262-263).

Reviewer’s comment:

L 402-406 We found that gerbils emitted nine types of vocalizations during the resident-intruder test. The spectral shapes of these calls were basically consistent with those reported in previous studies [17, 18]. Alert calls described in the same study [17] were not observed in our resident-intruder tests. The nine calls are categorized into two types; high-frequency vocalization and low-frequency vocalization with harmonics.

Nine types of vocalizations (L 402) cannot be categorized to the two call types. Using consistently own terms throughout the text is important for understanding the content by the readers. Replace with "The nine call types are categorized into two groups"

Author’s response: We replaced it (L405-406).

Reviewer’s comment:

L 406 low-frequency vocalization with harmonics.

Delete "with harmonics". Presence of visible harmonics is not exclusively attributive to low-frequency vocalizations, but depends on the size of the used for analysis spectral window.

Author’s response: We deleted "with harmonics" (L406).

Reviewer’s comment:

L 444 resident intruder

Replace with resident-intruder

Author’s response: We replaced it (L444).

Reviewer’s comment:

L 443-456 This paragraph do not provides useful information and is very far from the results of this study. Delete it.

Author’s response: We think that these discussions are useful to consider the multiple aspects of behaviors observed during agonistic interactions. We also rewrote the paragraph to make this more related to the present study.

Reviewer’s comment:

L 464-467 However, given that low-frequency calls were recorded even during the period before the resident started attacking the intruder in MF-VF pairs (this was not observed in the other groups), it would be reasonable to estimate that the majority of low-frequency calls were emitted by MF residents.

This not an argument. Avoid speculations. In rodents, a lot of information comes from olfactory channel. It is easy to propose that VF-intruders placed on territory of MF-residents, perceive the smell of adult females and start calling of fear before the first aggressive interaction. A few more opposite arguments can be advanced.

Author’s response: We made the statement moderate. We think that estimations/assumptions belong to the discussion and there is no digital separation between estimations (or rational speculations) and argument.

Reviewer’s comment:

L 473-476 However, it remains unknown why MF gerbils (or MF-VF pair) emitted more low-frequency calls compared with those in the other three pairs. We hypothesize that vocalizations are influenced by hormonal changes induced by parenting experiences in female Mongolian gerbils.

In the preceding paragraph, you write that it is impossible to establish, who is calling the low-frequency calls, MF or VF. However, here you attribute the calls to one of females of the pair. Please re-write to make the content consistent with the text above.

Author’s response: We rewrote that (L474).

 

Reviewer #2

Thank you for your kind suggestion to improve this manuscript.

As detailed below, we have addressed the reviewer’s reservation.

Reviewer’s comment: I think the descriptions of several behaviors that were scored should be more specific and detailed. We cannot define a target behavior with itself, for example, dig was defined by digging, and what is dig?.

Author’s response: We updated the descriptions in the table.

---

## [Decision Letter · Decision Letter 2]

20 Jul 2022

Vocalization during agonistic encounter in Mongolian gerbils: impact of sexual experience.

PONE-D-21-38093R2

Dear Dr. Yamamoto,

We’re pleased to inform you that your manuscript has been judged scientifically suitable for publication and will be formally accepted for publication once it meets all outstanding technical requirements.

Kind regards,

George Vousden

Staff Editor

PLOS ONE

Additional Editor Comments (optional):

Please accept our apologies for the delay in processing this decision.

Reviewers' comments:

Reviewer's Responses to Questions

**Comments to the Author**

1. If the authors have adequately addressed your comments raised in a previous round of review and you feel that this manuscript is now acceptable for publication, you may indicate that here to bypass the “Comments to the Author” section, enter your conflict of interest statement in the “Confidential to Editor” section, and submit your "Accept" recommendation.

Reviewer #1: (No Response)

Reviewer #2: All comments have been addressed

2. Is the manuscript technically sound, and do the data support the conclusions?

Reviewer #1: Yes

Reviewer #2: Yes

3. Has the statistical analysis been performed appropriately and rigorously? 

Reviewer #1: Yes

Reviewer #2: Yes

4. Have the authors made all data underlying the findings in their manuscript fully available?

Reviewer #1: Yes

Reviewer #2: Yes

5. Is the manuscript presented in an intelligible fashion and written in standard English?

Reviewer #1: Yes

Reviewer #2: Yes

6. Review Comments to the Author

Reviewer #1: The authors addressed all my comments.

I suggest minor changes, consistent to previous corrections.

L. 259 Delete “and multi-harmonics”

L. 264 Replace “syllable” with “group”

L. 446 Please corrects the typo; replace “repertories” with “repertoires”

L. 481 Consider replacing "Acknowledgment" with "Acknowledgement"

Reviewer #2: no major recommendation, but Ethical Note need to be added the Ethical Inspection License No:XXX,if you have.

7. PLOS authors have the option to publish the peer review history of their article (what does this mean?). If published, this will include your full peer review and any attached files.

Reviewer #1: No

Reviewer #2: No

---

## [Editor Report · Acceptance letter]

25 Jul 2022

PONE-D-21-38093R2 

Vocalization during agonistic encounter in Mongolian gerbils: impact of sexual experience. 

Dear Dr. Yamamoto:

I'm pleased to inform you that your manuscript has been deemed suitable for publication in PLOS ONE. Congratulations! Your manuscript is now with our production department. 

Kind regards, 

on behalf of

Dr. George Vousden 

Staff Editor

PLOS ONE